# Interpretable single-cell factor decomposition using sciRED

Delaram Pouyabahar [1,2], Tallulah Andrews [3,4,9] & Gary D. Bader [1,2,5,6,7,8,9] ✉

Single-cell RNA sequencing maps gene expression heterogeneity within a tissue. However, identifying biological signals in this data is challenging due to confounding technical factors, sparsity, and high dimensionality. Data factorization methods address this by separating and identifying signals in the data, such as gene expression programs, but the resulting factors must be manually interpreted. We developed Single-Cell Interpretable REsidual Decomposition (sciRED) to improve the interpretation of scRNA-seq factor analysis. sciRED removes known confounding effects, uses rotations to improve factor interpretability, maps factors to known covariates, identifies unexplained factors that may capture hidden biological phenomena, and determines the genes and biological processes represented by the resulting factors. We apply sciRED to multiple scRNA-seq datasets and identify sex-specific variation in a kidney map, discern strong and weak immune stimulation signals in a PBMC dataset, reduce ambient RNA contamination in a rat liver atlas to help identify strain variation and reveal rare cell type signatures and anatomical zonation gene programs in a healthy human liver map. These demonstrate that sciRED is useful in characterizing diverse biological signals within scRNA-seq data.

Single-cell RNA sequencing (scRNA-seq) maps heterogeneity in gene expression in large cell populations. This heterogeneity can be attributed to various factors, both observed and hidden[1]. We can broadly categorize sources of this heterogeneity into sample-level factors, such as experimental conditions or age and weight of patients, cell-level covariates like cell type identity, cell cycle stage, and library size, or as gene-level attributes such as pathways. Each of these categories can be further classified based on their biological or technical nature and whether they are known during experimental design, or observed in or inferred based on the data. Despite their importance, identifying and interpreting factors in scRNA-seq data remains challenging due to its noise, sparsity, and high dimensionality[2].

Matrix factorization (or decomposition) can identify multiple factors (signals) in a cell by gene scRNA-seq data matrix, each one capturing a unique pattern of covarying gene expression values that may represent a covariate, such as a set of genes affected by batch or a biological gene expression program[3]. Many factorization methods exist[4–13], however, they generally only identify factors and leave interpretation up to the user. Some recent methods, such as scLVM[14], fscLVM[15], and Spectra[16], can automatically interpret factors by matching them to pre-annotated gene sets (e.g. pathways), but do not consider a wider range of covariates important in single cell genomics data, such as sample-level attributes. To address this challenge, we developed Single-Cell Interpretable REsidual Decomposition (sciRED) to aid in scRNAseq factor analysis and interpretation.

sciRED enables factor discovery and interpretation in the context of known covariates and biological gene expression programs. sciRED provides an intuitive visualization of the associations between factors

[1]Department of Molecular Genetics, University of Toronto, Toronto, ON, Canada. [2]The Donnelly Centre, University of Toronto, Toronto, ON, Canada. [3]Department of Biochemistry, Schulich School of Medicine and Dentistry, University of Western Ontario, London, ON, Canada. [4]Department of Computer Science, University of Western Ontario, London, ON, Canada. [5]Department of Computer Science, University of Toronto, Toronto, ON, Canada. [6]Lunenfeld-Tanenbaum Research Institute, Toronto, ON, Canada. [7]Princess Margaret Research Institute, University Health Network, Toronto, ON, Canada. [8]CIFAR Multiscale Human Program, CIFAR, Toronto, ON, Canada. [9]These authors jointly supervised this work: Tallulah Andrews, Gary D. Bader. ✉e-mail: gary.bader@utoronto.ca

and covariates, along with a set of interpretability metrics for all factors. These metrics identify clear factor-covariate pairs as well as factors not matching known covariates but which are potentially interpretable. Factor-correlated genes and pathways also aid in interpretation. We apply sciRED to diverse datasets including the scMixology[17] benchmark and four biological single-cell atlases that contain known factors. We showcase its application in identifying cell identity programs and sex-specific variation in a kidney map, discerning strong and weak immune stimulation signals in a PBMC dataset, reducing ambient RNA contamination in a rat liver atlas to unveil strain-related variation, and revealing hidden biology represented by rare cell type signatures and anatomical zonation gene programs in a healthy human liver map. These demonstrate the utility of sciRED for characterizing diverse biological signals within scRNA-seq datasets.

## Results

To improve the efficiency and interpretability of matrix decomposition of single-cell RNAseq data, sciRED uses four steps to generate and characterize the resulting factors. These steps are: preprocessing data and applying matrix decomposition, evaluating factor-covariate relationships, examining unexplained factors, and determining biological interpretation of selected factors.

sciRED begins with the input cell by gene matrix and proceeds to: (1) Remove known confounding effects, factorize the residual matrix to identify additional factors not accounted for by confounding effects, and use rotations to maximize factor interpretability; (2) Automatically match factors with covariates of interest; (3) evaluate unexplained factors that may indicate hidden biological phenomena; (4) Determine the genes and biological processes represented by factors of interest (Fig. 1A). In the factor discovery phase, sciRED removes user-defined unwanted technical factors, such as library size and sample or protocol, as covariates within a Poisson generalized linear model (GLM)[18]. This regresses out the covariates and produces Pearson residuals. These residuals are then factored using Principal Component Analysis (PCA)[5] with varimax rotation[19] to enhance interpretability (Fig. 1B). To identify factors that explain a specific covariate, sciRED attempts to find matching labels (e.g. cell types, covariates of interest) for each factor using an ensemble classifier (see methods). Each cell is represented as a vector of factor weights, which are classified to predict covariate labels (e.g. "female" or "male" factors in the covariate "biological sex") using four machine learning classifiers (logistic regression[20], linear classifier/area under the curve (AUC)[21,22], decision tree[23], and extreme gradient boosting (XGB)[24]). Each factor is a feature and feature selection across an ensemble of these classifiers generates factor-covariate-level association (FCA) scores, which are visualized as a heatmap. To identify high-scoring factor-covariate pairs, we use three approaches. First, we use visual inspection of the FCA heatmap. Second, we compare the association scores of each factor to the background distribution of values in the FCA table to automatically highlight significant associations (see Methods, Fig. 1C). Third, we calculate a specificity measure to determine whether an explained factor captures the gene expression program related to a unique covariate or is associated with multiple covariate levels simultaneously. Interpretation is often easier when a factor matches only one covariate level.

The third step of sciRED evaluates unexplained factors for potential interpretability using three types of metrics: separability, effect size, and homogeneity, which are presented as a factor-interpretability score (FIS) table (Figs. 1D, S1, see Methods). For unexplained factors that should be prioritized for follow-up exploration because they may contain a hidden signal, ideally, cells should be scored highly by the factor (measured by effect size), there should be two populations of cells, ones that score highly and ones that don't (bimodal distribution, measured by separability) and the potential new signal is expected to be strong in cells ranked highly by that factor,

while other signals (e.g., known technical covariates) should be evenly distributed across this ranking (measured by homogeneity). The fourth step evaluates the factor loadings (gene lists) for biological signals. This involves manually investigating top genes and enriched pathways associated with the factor (Fig. 1E).

### sciRED improves factor discovery

We evaluated sciRED's factor discovery alongside eight other factor analysis methods (PCA, ICA[8], NMF[6], scVI[25], Zinbwave[9], cNMF[12], scCoGAPs[13], and Spectra[16]) on two datasets: the scMixology[17] benchmark dataset and a stimulated PBMC dataset[26], representing distinct biological and technical contexts (Figs. S2-S5). The scMixology dataset includes scRNA-seq profiles from a mixture of three cancer cell lines (H2228, H1975, HCC827), sequenced across three batches with different library preparation platforms (Drop-seq, Celseq2, and 10x Genomics) representing controlled single-cell data with well-defined signals. In contrast, the stimulated PBMC dataset consists of 10x Genomics droplet-based scRNA-seq data from eight lupus patients, collected before and after a 6-hour treatment with interferon (IFN)-β, providing a real-world example of biological single-cell data.

Four metrics were used to compare performance across methods: (1) the number of entangled covariates (indicating covariates matched to multiple factors); (2) the number of factors split across multiple covariates; (3) the number of covariate levels without an associated factor; and (4) runtime. Lower values for all metrics indicate stronger performance.

sciRED outperformed other methods, with the exception of scCoGAPs, in minimizing both entangled covariates and factors distributed across multiple covariates in both datasets. All methods showed comparable results in associating covariate levels with factors, especially in capturing biological signals. However, scCoGAPs missed identifying the megakaryocyte cell type in the PBMC dataset. The runtime analysis highlights sciRED's scalability as among the top four fastest on scMixology and second fastest to PCA on the larger PBMC data set, while scCoGAPs and Zinbwave were particularly slow running over 35 hours on scMixology; and on PBMCs scCoGAPs required 139 hours, whereas Zinbwave crashed after 24 hours due to RAM limitations. All methods were run single-threaded with default parameter settings on a workstation with Intel 3.0 GHz Xeon E5-2687W chip and 64 GB RAM.

Notably, sciRED's runtime scales linearly with both cell and gene counts (Fig. S6). This was shown using a human lung transplant dataset[27] (over 108,000 cells) and subsampling of both cell and gene counts. Runtime was tested across various cell numbers (20 K, 40 K, 60 K, 80 K, and 100 K cells with genes fixed at 2000) and gene counts (500, 2000, and 5000 highly variable genes with cells fixed at 40 K).

### sciRED finds cell type identity programs and sex-specific processes in a human healthy kidney map

We applied sciRED to a single-cell map of healthy human kidneys (Fig. 2) obtained from 19 living donors, with nearly equal contributions from both females and males (10 female, 9 male)[28]. sciRED successfully identified cell type identity programs for diverse kidney cell types (Fig. 2A, B, Fig. S7, Source Data) as well as a factor representing sex-specific differences. For instance, factors F1 and F13 capture the identity programs of proximal tubules and endothelial cells, respectively, while F18 represents sex-specific differences (Fig. 2C). Factor F18 has a high association score with male and female covariates based on the FCA heatmap, indicating it has captured sex variation. Further evaluation of the distribution of factor F18 across different cell types reveals a cell-type dependent distribution of sex-related variation, with a strong representation in the proximal tubule cell type population (Fig. 2D). The FIS heatmap indicates that F18 is highly specific and shows a low homogeneity score across sex covariates, confirming that cells from male and female individuals are not uniformly distributed

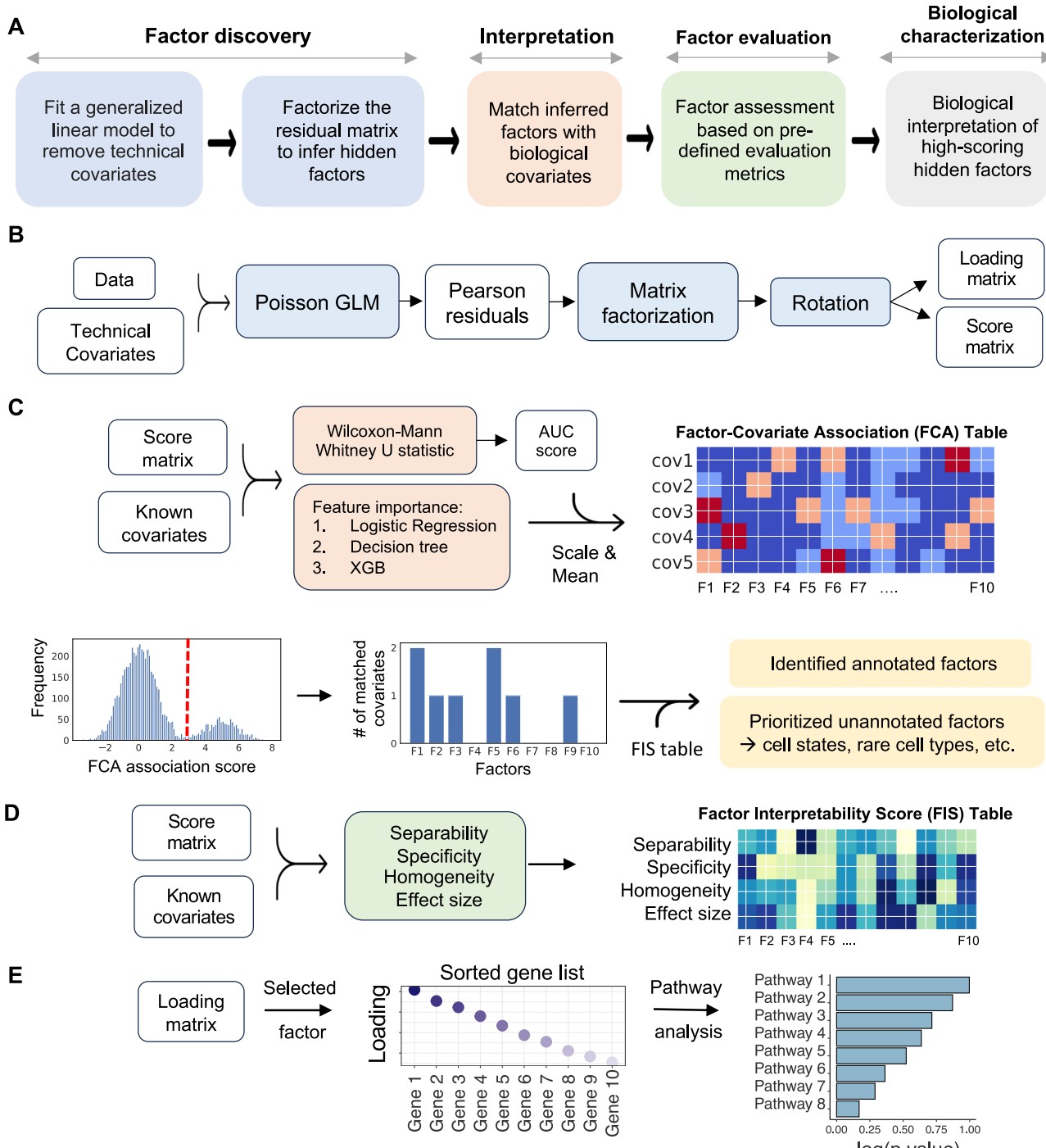

**Fig. 1 | sciRED overview. A** sciRED comprises four main steps: factor discovery, factor interpretation, factor evaluation and biological characterization. **B** In the factor discovery step, a Poisson generalized linear model is applied to the data to remove technical covariates, followed by extraction of residuals and factorization using PCA. The resulting score and loading matrices are then rotated for enhanced interpretability. The score matrix represents the projection of the original data onto the new factor space, illustrating the relationship between cells and factors. Each entry in this matrix reflects how much each cell contributes to the factors. The loading matrix contains the weights or coefficients that define the factors as linear combinations of the original genes. These weights can be used to rank genes according to their contribution to each factor, facilitating further interpretation of the factors. Models and algorithms are color-coded in blue, while input, output, and intermediate data elements are represented as white boxes. **C** Factor interpretation uses an ensemble classifier to match factors with given covariates, generating a Factor-Covariate Association (FCA) table. Covariate-matched factors are identified by thresholding FCA scores based on the distribution of all FCA scores. Unannotated factors may capture novel biological processes or other covariates. **D** Factor-interpretability scores (FIS) are computed for each factor. **E** The top genes and enriched pathways associated with a selected factor are identified for manual interpretation.

along F18 (Fig. 2E). Consistent with the original study, pathway analysis shows an increase in processes related to aerobic metabolism (such as aerobic respiration, oxidative phosphorylation, tricarboxylic acid (TCA) cycle, and electron transport chain) in males (Fig. 2F, G). These findings align with the higher basal respiration and ATP-linked respiration processes in males, as functionally validated in the original study[28]. This highlights sciRED's ability to identify cell type identity signatures and sample covariate-specific variation.

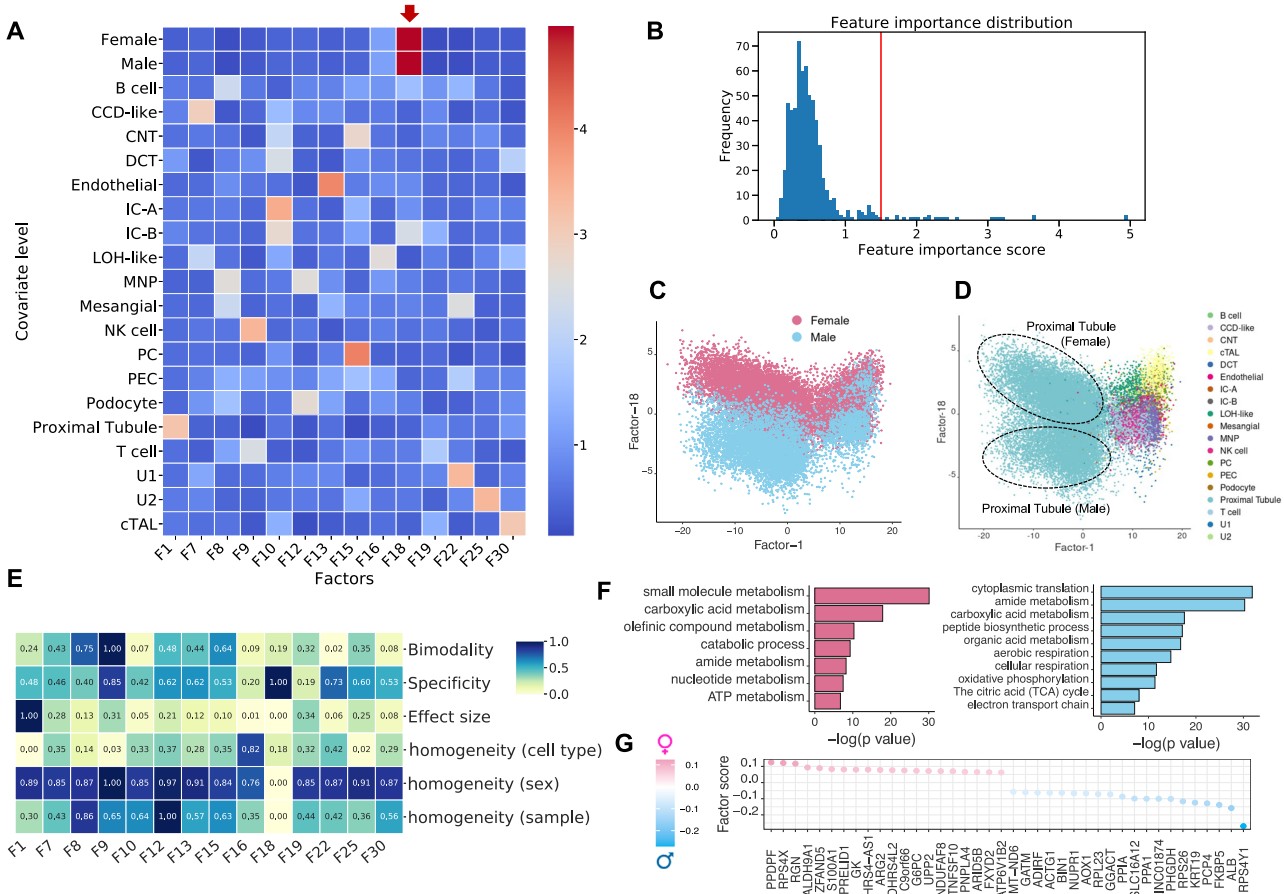

**Fig. 2 | Deciphering cell type identity programs and sex-specific processes in a human healthy kidney map.** We applied sciRED to a single-cell healthy human kidney map obtained from 19 living donors, with similar contributions from both females and males. **A** FCA heatmap with covariate levels as rows and associated factors as columns. Arrow highlights factor F18, which demonstrates a high association score with female and male covariates. **B** FCA score distribution for all factors. The red line indicates the automatically defined threshold used to identify

significant factor-covariate matches in the heatmap in (**A**). Distribution of cells over factor F1 and factor F18 colored based on **C** sex and **D** cell type covariates indicates the predominant representation of sex-related variation within the proximal tubule cell type. **E** FIS heatmap representing the interpretability scores of the selected factors. **F** Pathway analysis results on the top positive (female) and negative (male) loaded genes of factor F18. **G** Top 20 positively and negatively loaded genes of factor F18.

## sciRED identifies stimulation signals across lymphoid and myeloid cells in a stimulated PBMC dataset

We used sciRED to analyze a benchmark dataset comprising 10x Genomics droplet-based scRNA-seq PBMC data from eight lupus patients before and after a 6-hour treatment with interferon (IFN)-β[26] (Fig. 3). Our sciRED analysis successfully identifies cell type identity programs and stimulation-to-control axes of variation (Fig. 3A, Source Data). We identified two factors, F9 and F2, which capture stimulation signals (Fig. 3B). The FIS table indicates a high bimodality (separability) score for both factors, a high effect size for F2, and high specificity for F9 (Fig. 3C), illustrating the utility of the FIS table for capturing biological signals. Differential evaluation of the cell type representation within the two factors reveals a predominance of lymphoid and myeloid cells for F9 and F2, respectively (Fig. 3D). In particular, F9 represents a stronger overall stimulation signal and stimulation in lymphoid populations and F2 captures a stimulation signal in myeloid populations. Examination of the cell type distribution across F9 and F2 highlights distinct clustering between the non-stimulated control and stimulated groups along both factors (Fig. 3E, H). Immune response-related genes and biological processes, including interferon signaling, stress or pathogen response, and cytokine signaling, are enriched in both factors (Fig. 3F, G, I, J). This indicates sciRED's ability to identify both cell-type-specific gene expression activity programs, including stimulated cellular states.

## sciRED alleviates ambient RNA contamination for group-based comparison

We applied sciRED to a healthy rat liver atlas mapped in Dark Agouti (DA) and Lewis (LEW) strains[29]. This atlas contains hepatocyte-derived ambient RNA contamination, a known artifact likely caused by fragile hepatocytes leaking RNA into the cell homogenate before sequencing[30]. sciRED identified factors with cell type identities, as expected, along with two factors capturing strain-associated variation (F6 and F20) (Fig. 4A–F and S8A, Source Data). The FIS table shows that factor F6 has high specificity and separability, as well as low strain homogeneity (Fig. 4B). Factor F20 is the second strongest factor following F6 (Fig. 4B, C). F6 and F20 represent strain differences within the hepatocyte (Fig. 4D) and myeloid cell types (Figs. 4E, F, S8B and C), respectively. Standard differential expression analysis is not able to identify this signal due to strong ambient RNA contamination[29] (Fig. 4G). For instance, four hepatocyte genes—*Fabp1*, *Tmsb4x*, *Fth1*, *Apoc1*—are among the top differentially expressed genes within the myeloid cell type of both DA and LEW strains and are estimated to be ambient RNA by SoupX[31] (Fig. 4G, Source Data, see Methods). However, the top 50 myeloid strain-associated genes identified by sciRED F20 are free of such contaminants (Fig. 4H, Source Data). These myeloid-specific strain variations were experimentally validated in the original study. This shows the utility of sciRED in deconvolving biological signals from contamination to facilitate factor interpretation and group-based comparisons.

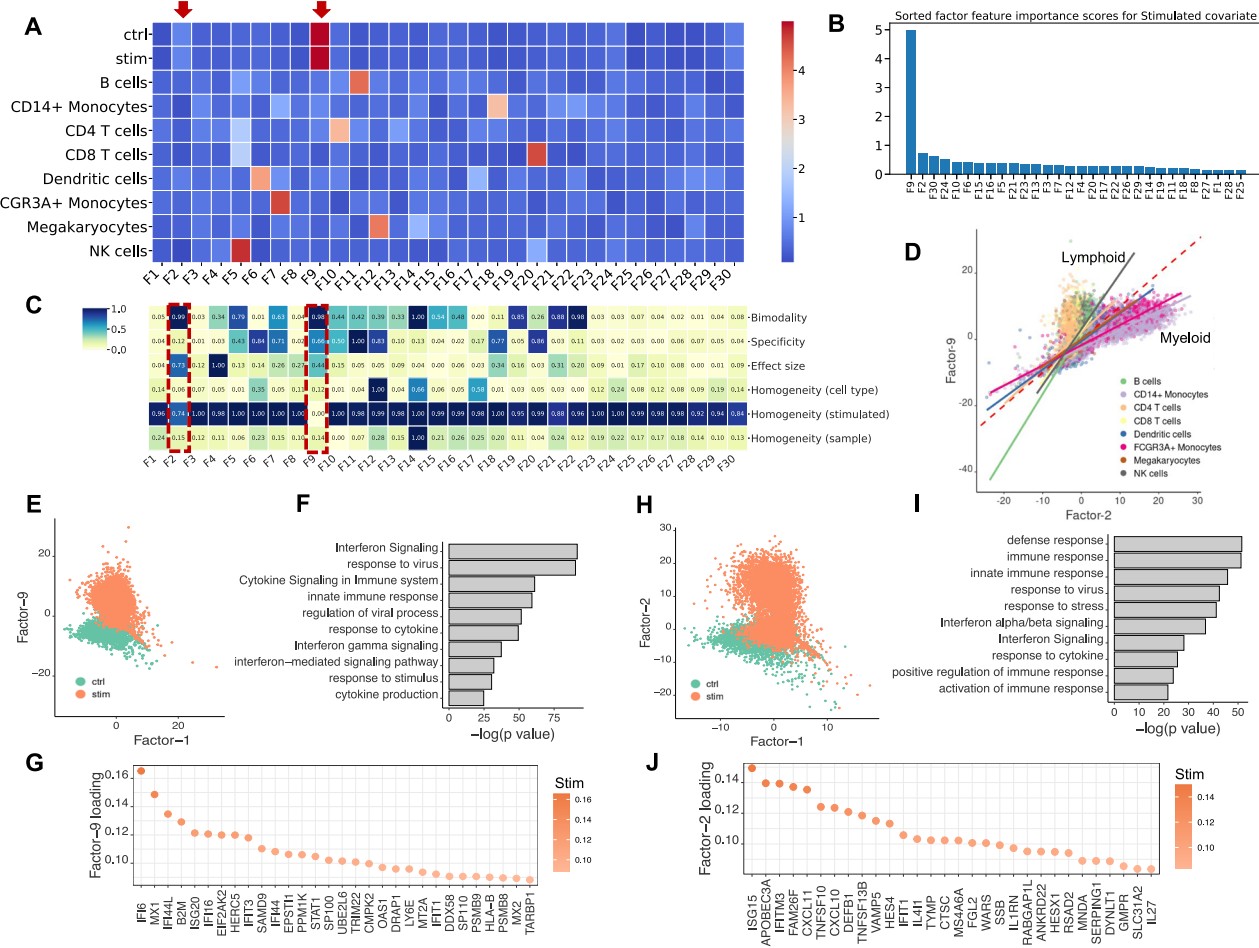

**Fig. 3 | Identifying interferon-β stimulation signals across lymphoid and myeloid cells in a PBMC dataset.** We used sciRED to analyze PBMC scRNA-seq from eight lupus patients before and after an interferon-β (IFN-β) treatment. **A** FCA heatmap displaying covariate levels as rows and associated factors as columns. Arrows highlight factors F9 and F2, which capture stimulation signals. **B** Sorted factors based on FCA score values for the stimulated covariate level. F9 and F2 are the top factors associated with the stimulation covariate. **C** FIS heatmap illustrating the interpretability scores of the selected factors. Red boxes highlight factors capturing IFN-β stimulation. **D** Cell distribution over factors F9 and F2, colored based on cell type covariates. The red dashed line represents the diagonal line

passing through the origin. Solid lines are the regression lines that fit each cell type. Lines with larger slopes than the diagonal represent cell types with higher association with F9, while lines with smaller slopes represent cell types more associated with F2. **E** Distribution of cells over factors F9 and F1 colored based on stimulation/control covariates, revealing distinct clustering between control and stimulated groups along the F9 axis. **F** Pathway analysis based on the top-loaded genes of factor F9. **G** Top 30 positively loaded genes of factor F9. **H** Distribution of cells over factors F1 and F2 colored by simulation state covariate ("stimulated" or "control"/ non-stimulated). **I** Pathway analyses based on the top-loaded genes of factor F2. **J** Top 30 positively loaded genes of factor F2.

## Exploring hidden biology in the healthy human liver atlas captured by unannotated factors

We applied sciRED to a healthy human liver atlas[32] and explored the unannotated biological signals (Fig. S9, Source Data). sciRED's FCA heatmap shows that most signals correspond to liver cell type identity programs (Fig. 5A). For example, F3 most strongly captures the cell identity of liver sinusoidal endothelial cells (LSECs), while F4 captures non-inflammatory macrophages. The FCA heatmap highlights nine factors (1, 10, 19, 20, 22, 26, 28, 29, 30) that are not associated with annotated cell types (Fig. 5B). To discern whether these factors represent technical or biological signals, we calculated the correlation between each factor and three major technical covariates—library size, number of expressed genes, and percentage of mitochondrial gene expression—as well as cell cycle (S and G2M) phase scores (Fig. 5C). Out of the nine factors, F1, F20, and F22 are correlated with technical covariates (R > 0.45). The remaining factors include five (F10, F19, F26, F28, and F30) that may represent unannotated biological signals, as well as F29, which shows a strong correlation with the cell cycle signature, indicating it might have captured a proliferative cell population. Evaluating the FIS table reveals that, among the six factors analyzed, all

except F19 are well-mixed based on the sample covariate, suggesting a possible sample-specific effect for F19. F10 and F29 stand out with higher bimodality scores, indicating they might more effectively separate a subpopulation of cells. F26 and F28 factors exhibit a low effect size, indicating weak signals (Fig. 5D). Factor F10 exhibits significant enrichment of a subpopulation of cells within the cholangiocyte cluster (Fig. 5E). The top loaded genes within this factor include *MUC5A, MUC1, TFF1, LGALS2, SLPI, TFF2, TFF3, KRT19, LGALS4,* and *PIGR*. The enrichment of biological processes such as ion export, regulation of transport activity, localization, and response to ER stress strongly suggests that F10 has captured the cellular program of a rare population of mucus-producing cholangiocytes that was not identified in the original study but was reported in a more comprehensive subsequent human liver single cell map[33]. Factor F30 is enriched within a subpopulation of cells labeled as plasma cells in the original map (Fig. 5F). The top genes in the loading vector include IgK+IgG+ B cell marker genes, including *IGHG4, IGHG1, IGHG3, IGKV1-12, HERPUD1, IGKV1-18, IGKC, SSR4, IGKV3-20,* and *IGKV3-21* (Fig. 5F). Pathway enrichment analysis highlights biological processes such as protein folding and maturation, antibody-mediated complement activation, protein transport, and ER to cytosol transport

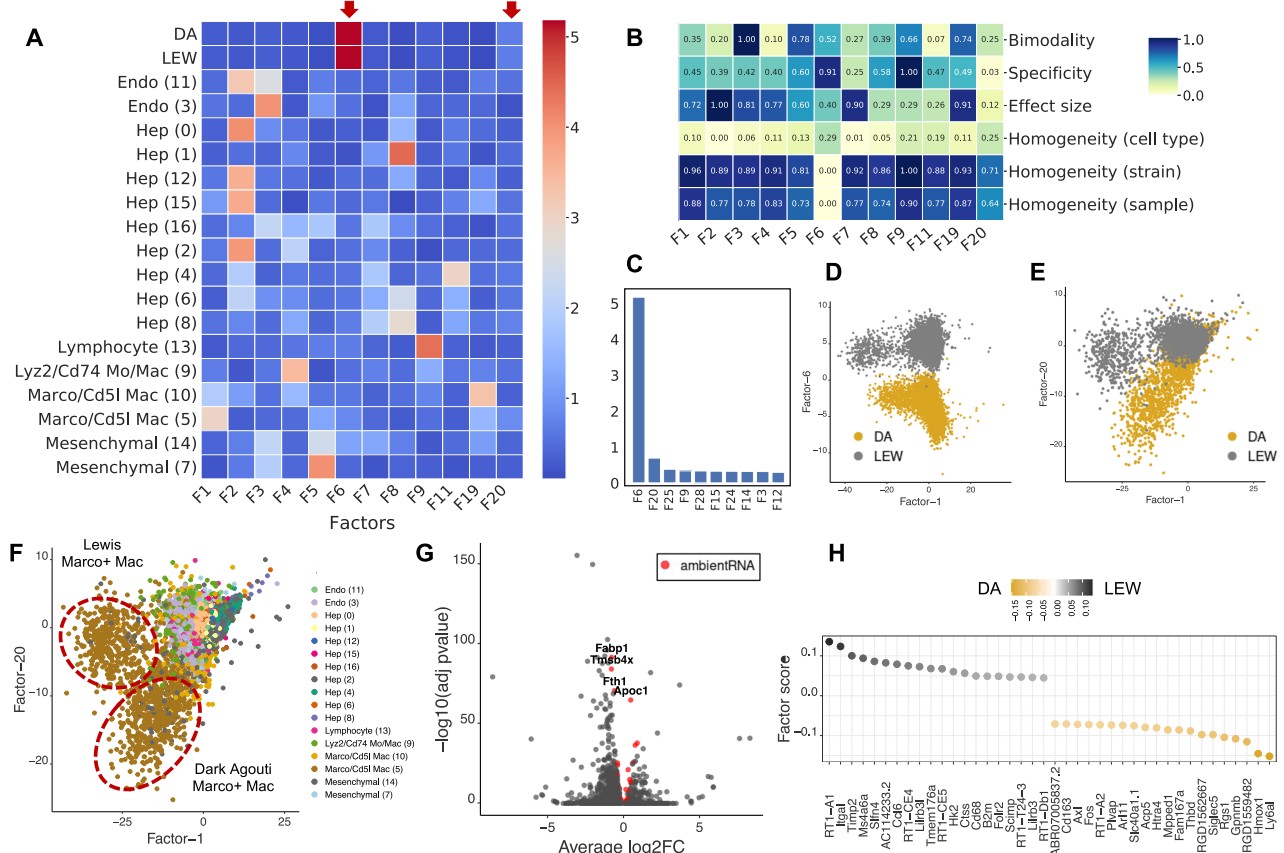

**Fig. 4 | sciRED identifies strain-based variation despite ambient RNA contamination in a rat liver map.** We applied sciRED to a healthy atlas of the rat liver from two rat strains, Dark Agouti (DA) and Lewis (LEW) containing hepatocyte-derived ambient RNA contamination. **A** FCA heatmap displaying covariate levels as rows and associated factors as columns. **B** FIS heatmap illustrating the interpretability scores of the selected factors. **C** Factors F6 and F20 are most associated with strain variation. **D** Distribution of cells over factors F6 vs. F1 colored by strain and **E** factors F20 vs. F1 colored by strain and **F** by cell type, indicating that factor F20 captures strain variation within the myeloid population. **G** Volcano plot of differential expression between strains within myeloid cells. Red dots are hepatocyte-derived ambient RNA transcripts as estimated by SoupX. Four hepatocyte genes—*Fabp1*, *Tmsb4x*, *Fth1*, and *Apoc1*—are labeled among the top differentially expressed genes within the myeloid cell type of both DA and LEW strains. These genes are among the top 50 ambient RNA transcripts derived from SoupX in all four rat liver samples (see Source Data). **H** Top 50 myeloid strain-associated genes identified by sciRED factor 20, free of contamination from hepatocyte-derived ambient RNA.

(Fig. 5F). These findings suggest that F30 may represent an antibody-secreting IgK+IgG+ B cell[33] gene expression program, not described in the original study due to its low frequency, but later captured in an expanded human liver single-cell map[33]. Factor F19 is enriched in two hepatocyte clusters, Hep1 and Hep2, which were annotated as peri-centrally zoned hepatocytes in the original study (Fig. 5G). The presence of pericentral markers such as *CYP3A4*, *GLUL*, and *OAT*, along with enrichment in biological processes such as xenobiotic metabolic processes, small molecule metabolism, and lipid and fatty acid metabolism, suggests that F19 captures the anatomical pericentral signature within the hepatic lobules[30,32,34](Fig. 5G). Factor F29 is correlated with cell cycle signatures and inversely enriched within a population of γδ T cells (Fig. 5H). The top loaded genes for F29 include cell-cycle-related regulatory genes *UBE2C*[35], *TOP2A*[36], *KPNA2*[37], *CKS2*[38], and *BIRC5*[39], and pathway analysis reveals enrichment in cell cycle and cell division, RNA splicing and processing, and nuclear division (Fig. 5H). Together, these results suggest that this factor captures the cell cycle state within the γδ T cell subset. We could not clearly interpret factors F26 and F28. In conclusion, sciRED is able to identify weak signals, such as rare cell types and subtle cell states, that were overlooked by standard single-cell analysis pipelines in the original study.

## Discussion

We developed sciRED, a novel single-cell transcriptomics data interpretation method that combines unsupervised factor analysis and supervised covariate modeling to identify biological and technical signals within single-cell transcriptomics data. We introduce new metrics to assess factor interpretability to help characterize known and unknown sources of variation in the data. We also showed that regressing out known technical factors before factorization aids in data interpretation.

To test sciRED, we analyzed a series of datasets with diverse known covariates and showed that sciRED could recover factors associated with these covariates. This works well, but may miss uninterpreted signals in the original data. Although many single-cell transcriptomics simulation tools exist[9,40–45] to assist in benchmarking analysis methods, there has been little focus on factor simulation. We simulated factors to evaluate interpretability metrics, but our approach was simplistic and can be improved. sciRED uses PCA coupled with varimax rotation as a high-performing matrix decomposition method, though many other factor analysis methods exist[46]. Our benchmarking results on the controlled scMixology dataset and a biologically complex PBMC dataset indicate that sciRED generally outperforms other methods in terms of combined interpretability and runtime efficiency. Tools dedicated to simulating loading and score matrices from single-cell transcriptomics, ranging from simple to complex structures, could aid in benchmarking the performance of matrix factorization methods across different scenarios. Such tools could also help answer questions such as determining the minimal sample size required to recover a factor with weak loadings. These

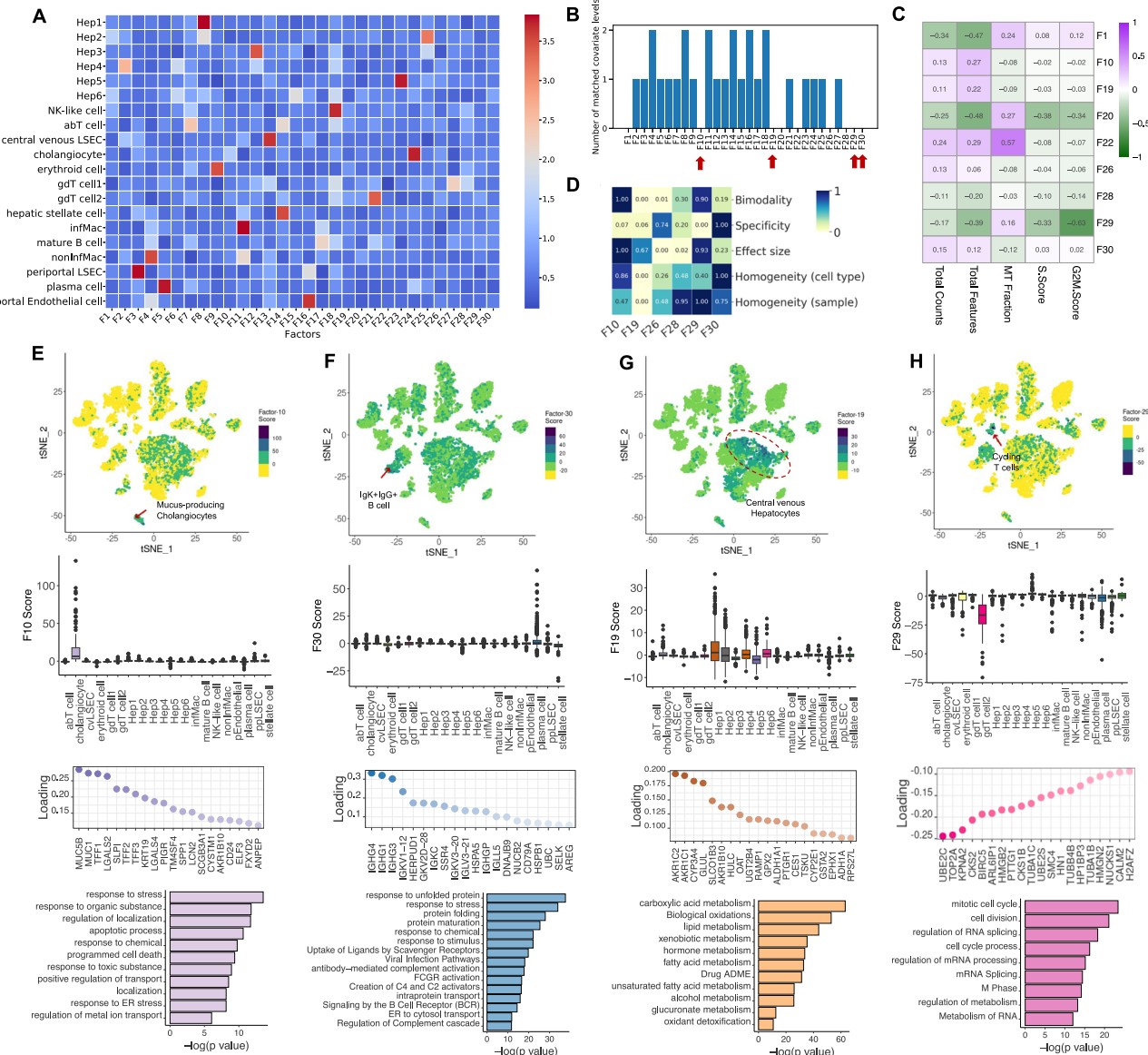

**Fig. 5 | Exploring hidden biology in the healthy human liver atlas using unannotated factors.** We demonstrate how sciRED facilitates the identification of hidden biology in a healthy human liver single-cell transcriptomic atlas. **A** sciRED's FCA heatmap reveals signals corresponding to human liver cell type identity gene expression programs. **B** Distribution of the number of matched covariate levels per factor identifies nine unannotated factors (F1, F10, F19, F20, F22, F26, F28, F29, F30) not associated with any given covariate. **C** Correlation analysis between unannotated factors and technical covariates (library size, number of expressed genes, percentage of mitochondrial gene expression) and cell cycle (S and G2M) phase scores. **D** FIS heatmap indicating the interpretability scores of unexplained factors uncorrelated with technical covariates. E–H The first row shows the distribution of selected factors on the atlas tSNE plot, where each dot represents a cell, and colors indicate factor score values. The second row presents boxplots of factor scores across cell types, with the box representing the interquartile range (IQR), the line indicating the median, whiskers extending to 1.5 × IQR, and dots representing

individual cell factor scores identified as outliers. The third row displays the top-loaded genes, and the fourth row provides pathway analysis for factors (**E**) F10, (**F**) F30, (**G**) F19, and **H**) F29. **E** Factor 10 exhibits significant enrichment within a subpopulation of cells within the cholangiocyte cluster, suggesting the capture of a rare population of mucus-producing cholangiocytes. **F** Factor 30 demonstrates positive enrichment within a subpopulation of cells labeled as plasma cells in the original map, suggesting that it captures antibody-secreting IgK+IgG+ B cells. G Factor 19 is positively enriched in pericentrally zoned hepatocytes (clusters Hep1 and Hep2), capturing an anatomical pericentral gene expression signature within hepatic lobules. H Factor 29 is correlated with cell cycle signatures and inversely enriched within a population of γδ T cells, suggesting capture of cell cycle state within this T cell subset. Pathway enrichment analysis of the top 200 genes from ordered loadings was performed using gProfiler in "ranked" mode with default Gene Ontology Biological Process and Reactome databases.

simulation methods could draw inspiration from factor simulation techniques used in psychometric studies[47–50].

We present a suite of metrics designed to evaluate the interpretability of factors derived from a matrix factorization method within a single dataset. sciRED's modular design enables flexible integration with external data processing and factorization methods. Preprocessed data can enter directly at the factor identification stage, and

previously derived factors and covariates can be imported for factor-covariate association, bypassing the initial steps if matrix factorization has already been applied using another method. Both FCA and FIS heatmaps can provide guidance for interpreting factors derived from other matrix factorization methods (Fig. S10). These metrics help identify which of the K factors generated by a given factorization algorithm are likely to be interpretable. However, factor distributions

can exhibit various patterns that the provided metrics may not fully capture. For unexplained factors, even one high metric value can be sufficient to highlight them as potentially interesting. In such cases, further interpretation should rely on enriched genes and pathways. Moreover, the optimal factorization method and its specific settings—such as the number of genes included—may vary depending on the input dataset. Variables like sample complexity (including diversity of cell types, presence of rare populations, and number of covariates in the experimental design), signal-to-noise ratio (including contamination level and dropout rate), number of batches, and variation in dissociation protocols and capture technologies in multi-sample studies can influence the choice of factorization method. Developing metrics to compare and identify the optimal factor analysis method and its parameter settings to generally improve interpretability across different methods would be beneficial but remains a challenge.

In sciRED we focused on matrix factorization. However, a limitation of this approach is the assumption that cells can be represented as linear combinations of gene expression signatures, which may limit our ability to capture non-linear patterns. Alternative approaches, such as deep learning-based latent variable models like variational autoencoders (VAEs), can incorporate non-linearities and interactions between latent variables[25] in the factorization step. While the interpretability measures we developed would be applicable to VAE-derived latent spaces, non-linear patterns may inherently be more challenging to interpret. Thus, there remains a trade-off between interpretability and the ability to model complex biological phenomena effectively. The proportion of cellular programs that can be effectively captured by linear versus non-linear approaches remains uncertain for any given dataset. Enhancing the interpretability of deep learning-based models, such as VAEs, while maintaining their ability to capture complex biological phenomena is an active research area[51,52].

We only consider applying sciRED to scRNAseq data. However, it could be applied to bulk RNAseq datasets with tens to hundreds of samples. We do not recommend using the Poisson GLM step in sciRED for full-length, plate-based datasets such as Smart-seq2, as these data typically follow a different statistical distribution compared to UMI-based data[53,54]. However, users can apply their custom preprocessing approach and then input these data into sciRED's factor identification which would allow sciRED to be used on Smart-seq2 data. sciRED could also be extended to function with spatial and multi-omics data with multi-sample datasets, appropriate factor analysis methods and a good understanding of how technical variation affects these data. Analysis of one spatial transcriptomics dataset shows that sciRED can identify spatially distinct gene expression patterns within complex tissue structures (Figs. S11 and S12). However, further refinement may be needed to enhance its handling of spatial data and other emerging data types, particularly in terms of selecting appropriate error models, and optimizing the metrics used to assess factor quality across diverse experimental technologies. We expect that as single-cell and multi-omics technologies evolve, integrating and comparing factors extracted from many datasets and various data modalities will deepen our understanding of cellular systems in healthy and diseased tissue.

## Methods

### sciRED factor discovery framework

We model the read counts for n cells and g genes (i.e., $Y_{n \times g}$) as a combination of annotated and unannotated factors as follows:

$$Y_{n \times g} = C_{n \times p}\beta_{p \times g} + F_{n \times f}A_{f \times g} + U_{n \times u}H_{u \times g} + \varepsilon_{n \times g}$$

Where:

$Y_{n \times g}$ represents the observed data matrix, typically with dimensions $n \times g$ (where $n$ is the number of samples or cells and $g$ is the number of variables or genes)

$C_{n \times p}$ is the matrix of technical covariates with dimensions $n \times p$ (where $p$ is the number of covariates), such as library size, batch ID, cell cycle stage

$\beta_{p \times g}$ represents the coefficient matrix for technical covariates with dimensions $p \times g$

$F_{n \times f}$ is the matrix of annotated factors with dimensions $n \times f$ (where $f$ is the number of factors)

$A_{f \times g}$ is the matrix of loadings for annotated factors with dimensions $f \times g$

$U_{n \times u}$ is the matrix of unannotated factors with dimensions $n \times u$ (where $u$ is the number of unannotated factors)

$H_{u \times g}$ represents the matrix of loadings for unannotated factors with dimensions $u \times g$

$\varepsilon_{n \times g}$ is the error term, with dimensions $n \times g$

Annotated factors are those matched with known (given) covariates as indicated in the factor-covariate association (FCA) table. The solution to the above equation is reached through a two-step process.

1. Derivation of residuals

The first step aims to remove the effect of technical covariates from the data.

Input counts, represented by the matrix $Y_{n \times g}$ are modeled as a Poisson generalized linear model (GLM) to account for the matrix's distributional properties[11,55,57]. Technical covariates $C_{n \times p}$ are incorporated into the model to capture their effects on the count data. The statsmodels package[58] (version 0.11.0) is used for GLM implementation.

Pearson residuals from this model are computed as the difference between observed and predicted counts divided by the square root of the predicted counts:

$$r_i = \frac{y_i - \hat{y}_i}{\sqrt{VAR(\hat{y}_i)}}$$

where $VAR$ is the variance. For Poisson GLM, $VAR(\hat{y}_i) = \hat{y}_i$.

The Pearson residuals are used for subsequent matrix decomposition.

Two additional residual types were evaluated, but not used in sciRED. Response residuals represent the difference between the observed count and the predicted mean count for each observation.

Response Residuals: $resid_i = y_i - \hat{y}_i$

Deviance residuals represent the individual contributions of samples to the deviance $D$, calculated as the signed square roots of the unit deviances. Deviance Residuals:

$$d_i = sign(y_i - \hat{y}_i) \times \sqrt{2 \times (y_i \log(y_i/\hat{y}_i) - (y_i - \hat{y}_i))}$$

Pearson residuals are used for sciRED because they are established in other single cell analysis methods[55,56]. However, sciRED results are robust to the choice of residual (among Pearson, response, deviance) (See Methods section "Classifier ensemble").

2. Residual decomposition

After obtaining Pearson residuals from the count data, a matrix factorization technique is employed to uncover underlying patterns, including the annotated and unannotated factors. PCA factors are calculated using Singular Value Decomposition (SVD)[59] as implemented in scikit-learn[60] (version 0.22.1). Following factor decomposition, we apply varimax rotation to enhance the interpretability of the principal axes in the feature space. To achieve this, we reimplemented the optimized estimation procedure as described below.

### Rotation types

Factor analysis typically comprises two sequential stages. Initially, loadings are computed to best approximate the observed variances within the data. However, these initial loadings may lack interpretability; thus, we apply rotation to generate a revised set that is more

easily interpretable. There are two primary rotation types in factor analysis: orthogonal and oblique rotations. Orthogonal rotation, such as varimax, seeks to produce orthogonal factors with zero correlations. Intuitively, varimax rotation aims to identify factors associated with a limited number of variables, thereby it promotes the detection of distinct factors rather than those affecting all variables evenly. Mathematically, interpretability is achieved by maximizing the variance of the squared loadings along each principal component axis. Varimax rotation seeks to maximize the Kaiser criterion:

$$v = \sum_j v_j = \sum_j \left\{ \left[ g \sum_i \left( l_{ij}^2/h_i^2 \right)^2 - \left( \sum_i l_{ij}^2/h_i^2 \right)^2 \right] / g^2 \right\}$$

Where $v$ is the Kaiser criterion and $l_{ij}$ is the loading value of $i^{th}$ gene and $j^{th}$ factor, and $g$ is the total number of genes. The communality $h_i$ is calculated as the squared sum of the loading values for each gene.

$$h_i^2 = \sum_j l_{ij}^2$$

To explicitly specify the rotation matrix, we can reformulate the Kaiser criterion using the following notation. Let $L$ be a $g \times k$ loading matrix (eigenvectors), and $R$ denote a rotation matrix such that $R^T R = I_k$, where $I_k$ is the $k \times k$ identity matrix. Additionally, let $R_{ij}$ represent the scalar element in the $i$th row and $j$th column of matrix $R$. Varimax rotations can now be described as follows:

$$R_{varimax} = argmax_R \left( \sum_{j=1}^{k} \sum_{i=1}^{g} (LR)_{ij}^4 - \frac{1}{g} \sum_{j=1}^{k} \left( \sum_{i=1}^{g} (LR)_{ij}^2 \right)^2 \right)$$

where $R_{varimax}$ denotes the resulting rotation matrix.

The rotation matrix is computed using an iterative method relying on SVD to achieve sparsity in the loadings. Subsequently, the rotated loadings and score matrices are derived by multiplying the original loadings and score matrices with the rotation matrix, respectively. The optimization algorithm is elaborated on in detail in Appendix A of Stegmann et al.[61]. We re-implemented the base R varimax rotation function in Python.

Oblique rotations, such as Promax[62], allow factors to be correlated, thereby relaxing the orthogonality assumption[63]. This flexibility can be beneficial when factors are expected to be correlated within the underlying structure of the data. Promax initially applies the varimax method to generate a set of orthogonal results. Subsequently, it constructs an ideal oblique solution to exaggerate the orthogonal rotated gene-loading matrix. Finally, it rotates the orthogonal results to achieve a least squares fit with this ideal solution.

We define a pattern matrix $P = (p_{ij})$, as the following ($k > 1$):

$$p_{ij} = \frac{|l_{ij}^{k+1}|}{l_{ij}}$$

Each element of $P$ matrix is the $k^{th}$ power (typically 3rd power) of the corresponding element in the row-column normalized varimax loading matrix. Next, the least squares fit of the orthogonal matrix of factor loadings to the pattern matrix is computed.

$$R_{Promax} = argmin_R ||P - L_{rot}R||_2$$

$$R_{Promax} = (L'_{rot}L_{rot})^{-1} L'_{rot}P$$

where $R_{Promax}$ is the unnormalized rotation matrix, $L_{rot}$ is the varimax rotated loadings, and $P$ is the pattern matrix defined above. The columns of $R_{Promax}$ are normalized such that their sums of squares are equal to unity. We reimplemented the base R promax function in Python.

To evaluate the impact of factor rotations, we applied PCA, sciRED (varimax-based[19]), and promax-rotated[62] PCA to the Pearson residual of the scMixology dataset after regressing out protocol and library size. The factor-covariate association scores reveal a high correlation between sciRED and Promax. Both methods enhance specificity and achieve one-to-one association between factors and cell line covariates, outperforming unrotated PCA (Figure S13). We used varimax as the default rotation method for sciRED, as it has been established to perform well for interpretable factor analysis[64,65].

## Factor identification benchmarking

For both scMixology and PBMC datasets, PCA and ICA were applied to normalized (library-regressed) data, while NMF, scVI, Zinbwave, cNMF, and scCoGAPS were run on filtered raw counts. All methods used the top 2000 highly variable genes, following sciRED's standard pipeline. The number of factors (K) was set to 30 across all methods; for additional comparison, K was also set to 10 for the scMixology dataset.

Spectra was applied exclusively to the PBMC dataset, as it relies on the Cytopus knowledge base, which contains gene sets tailored for standard scRNA-seq data but lacks coverage for cancer cell line-specific gene sets. For PBMC, cell type labels were standardized to CD4-T, mono, cDC, NK, CD8-T, and B, with an "all-cells" category used for cell types without specific annotations. The number of factors per cell type was set to five to approximate the total number of factors used in the other methods.

All methods were run in single-threaded mode with default parameters on a workstation with Intel 3.0 GHz Xeon E5-2687W chip and 64 GB RAM.

## Data pre-processing and handling of batch effects with sciRED

sciRED takes raw count (not batch corrected) data as input, removing cells and genes with zero total read count and retaining only the top highly variable genes, typically the top 2000 as in standard scRNA-seq workflows. While using all genes is an option, the first step—fitting the Poisson GLM—becomes more time-consuming with larger gene sets; thus, a smaller set of highly variable genes is used. The minimum required covariate is library size (total read count per cell), which serves as an equivalent normalization step.

Additional covariates can be incorporated into the GLM model based on specific analysis goals; however, technical covariates should be removed with caution to prevent the unintentional exclusion of biologically relevant signals. For instance, in analyses focused on sex-specific variations, adjusting for sample IDs as covariates can inadvertently eliminate sex-associated signals if sample and sex effects are confounded.

In multi-sample analyses, we recommend merging all samples without batch correction. Batch correction is generally unnecessary with sciRED, as our analyses demonstrate robust batch effect handling without directly correcting the data matrix or latent embedding. To evaluate batch effect handling, sciRED was tested on two PBMC datasets profiled using 10x Genomics single-cell 3' and 5' gene expression libraries with strong batch effects (Fig. S14). sciRED was applied to the combined count matrices, with library and sample IDs regressed as covariates in the Poisson GLM step. The UMAP projection (Figures S14AB) reveals batch-related clustering without correction; however, sciRED analysis (Fig. S14C, D) shows well-integrated cell type identities, even across assay types. For instance, Factors F1 and F4, which identify CD14+ monocytes and B cells, respectively, are well integrated across both assays. The distributions of cells over F1 and F4, colored by cell type and assay, demonstrate effective batch integration, and the corresponding box plots and UMAP enrichment patterns further confirm these results (Fig. S14E–J).

## Impact of data sparsity on factor decomposition performance

To assess the impact of data sparsity on sciRED's decomposition, we systematically varied the sparsity level (0.01, 0.3, 0.5, 0.7, 0.9, 0.95, 0.99) in the human liver atlas dataset by incrementally replacing randomly selected gene expression values with zeros (Fig. S15). For each level, we evaluated the factors based on average and maximum variance explained, the percentage of matched factors, and the percentage of matched covariates. Our results indicate that the average variance explained by all factors remains largely robust across different sparsity levels, likely due to sciRED's rotation step, as this trend is not observed for PCA factors. The maximum variance explained by factors is also more stable with sciRED's rotated factors compared to PCA as sparsity increases. As expected, the percentage of matched factors decreases with rising sparsity, with a marked drop around a sparsity threshold of ~70%. At lower sparsity levels (0.01 and 0.5), the FCA heatmaps display a balanced distribution of matched covariates per factor, while at higher sparsity (0.95 and 0.99), most covariates align with only a few factors, reducing interpretability. However, the percentage of matched covariates remains stable, as most covariates continue to align with at least one factor.

## Impact of factor number (K) on decomposition results

To evaluate the impact of factor count (K) on sciRED's decomposition results, we tested various factor numbers—5, 10, 20, 30, and 50—on the human liver map (Fig. S16). At lower factor counts, such as $K = 5$, each factor showed higher covariate associations (e.g., Factor 1 was matched to nine covariates). As K increased, each factor aligned with fewer covariates, and the percentage of matched factors decreased. At $K = 50$, 54% of factors had no covariate matches. Correlating factors between the $K = 10$ and $K = 30$ decompositions revealed that individual factors in $K = 10$ (e.g., F1, F3, and F7) mapped to multiple factors in $K = 30$, suggesting that low factor limits may lead to each factor capturing multiple gene expression programs, complicating interpretation. Repeating this analysis on the PBMC dataset yielded similar results: as the number of factors increased, each factor aligned with fewer covariates, and a higher proportion of factors remained unmatched at higher K values (Fig. S17).

The ideal factor count varies by dataset and is affected by biological and technical signal diversity. While a Scree plot is commonly used in PCA to determine factor counts, sciRED's rotation step changes the ordering by variance explained, making this approach less effective. Empirically, a moderately high K (e.g., 30) has yielded interpretable results across datasets. We recommend starting with a higher K value and excluding factors with no covariate matches or low factor importance scores (FIS).

## Feature importance calculation

We evaluated a range of classifiers for inclusion in the sciRED ensemble classifier. Each classifier is trained on individual levels of each covariate separately (e.g., "female" and "male" for a "biological sex" covariate). Each classifier uses a different approach to estimate feature importance:

1. Logistic regression: feature importance is the magnitude of the coefficient for each factor, which represents the change in the log-odds of belonging to a covariate level per unit of factor weight.
2. Decision trees, random forest[66] and extreme gradient boosting: feature importance scores represent the decrease in covariate mixing (e.g. Gini impurity or entropy) when the feature is used within a tree averaged across all trees.
3. K-nearest neighbor[67] (KNN): feature importance is estimated as the decrease in predictive accuracy when the values for that feature are randomly permuted. This value is calculated as the average across five permutations for each factor based on the default scikit-learn package implementation.

4. Linear classifier (AUC): feature importance for a linear classifier (i.e. fixed threshold) is calculated as the area under (AUC) the receiver-operating characteristic (ROC) curve. The AUC for one-dimensional data is equivalent to the Wilcoxon or Mann-Whitney U test statistic with the relation:

$$AUC = U/(n_0 \times n_1)$$

Where U is the Mann-Whitney U statistic, and $n_0$ and $n_1$ are the sample sizes of the two groups being compared. The Mann-Whitney U test is a non-parametric test used to assess whether two independent samples are selected from populations having the same mean rank. Here, samples are defined as factor scores for the target group (cells labeled with the covariate level of interest) and the non-target group. In the context of feature importance, a higher AUC value indicates that the factor is better at separating the classes, while a lower AUC value suggests less discriminatory power. The scikit-learn package is used to implement decision tree, random forest, logistic regression and KNN classifiers with default parameters. XGBoost (version 1.5.0) and SciPy[68] (version 1.4.1) packages are used for XGB and the Mann-Whitney U test, respectively.

## Classifier ensemble

We optimized the sciRED classifier ensemble by evaluating different classifier combinations on four independent datasets: a healthy human kidney map, a healthy human liver map, a PBMC atlas, and the scMixology benchmark dataset (Fig. S18). For each experiment, we randomly shuffled the covariate labels to generate a null distribution of classifier association scores and calculated the average number of significant associated factors ($p < 0.05$) per covariate level (Fig. S18A). We defined a one-to-one association between factors and covariates as the optimally interpretable result. This analysis shows that the sciRED classifier (ensemble of logistic regression, linear classifier/area under the curve (AUC), decision tree, and extreme gradient boosting (XGB)) outperforms or matches the performance of the individual classifiers, depending on the data set. Initially, six classifiers—AUC, K-Nearest Neighbors (KNN), logistic regression, decision tree, random forest, and XGB—were compared on the scMixology benchmark dataset. Due to KNN's poor classification performance and random forest's inferior scalability, they were excluded from the sciRED ensemble model (Fig. S18B, C). Benchmarking on the four independent datasets described above demonstrates sciRED's superior performance relative to single classifiers (Fig. S18D–G).

To optimize the ensemble score, we also tested all combinations of three different scaling methods and two mean calculations (Fig. S19). Specifically, we considered standardization, min-max scaling, or rank-based scaling combined with arithmetic or geometric means[69]. By comparing scores for the real scMixology data vs. permuted data labels, we identified standardization followed by arithmetic mean calculation as the optimal scoring method (Fig. S19).

Standardization, min-max scaling, and rank-based scaling are defined as follows:

Given a set of feature importance scores for each factor $i$: $x = \{x_1, x_2, \ldots, x_n\}$, and their ascending order of ranks $r = \{r_1, r_2, \ldots, r_n\}$, where $n$ is the total number of features (factors):

$$standardized_i = \frac{x_i - \text{mean}(x)}{std(x)}$$

$$\text{min} - \text{max}\ scaled_i = \frac{x_i - \text{min}(x)}{\text{max}(x) - \text{min}(x)}$$

$$rank - scaled_i = \frac{r_i}{n}$$

Rank scaling generates a value between 0 and 1 to each data point based on its rank within the dataset. Lower values in the original dataset will have lower rank-based scaled values, while higher values will have higher rank-based scaled values.

Arithmetic and geometric means across classifiers are calculated as follows:

$$Arithmetic\ mean = \frac{1}{n}\sum_{i=1}^{n} x_i$$

Where n is the number of values.

$$Geometric\ mean = \left(\prod_{i=1}^{n} x_i\right)^{\frac{1}{n}}$$

The geometric mean is the $n^{th}$ root of the product of a set of values.

The effect of the choice of residual (Pearson, response, deviance) on sciRED performance was also evaluated, indicating that sciRED results are robust to the choice of residual (Fig. S20).

### Determining significant factor-covariate associations

FCA scores are binarized into significant and non-significant associations based on a threshold. This threshold is automatically determined using Otsu thresholding[70]. Otsu thresholding iterates through all potential threshold values and computes a separability measure in each iteration. The threshold value that maximizes this separability measure is chosen as the optimal threshold to partition the FCA score distributions into significant and non-significant associations. The scikit-image package[71] (version 0.23.2) is used for implementation.

### Benchmarking covariate-factor association metrics based on permutation results

We used a permutation test to evaluate the significance of each covariate-factor association. This process entailed randomly shuffling cell covariate labels 500 times and recalculating factor-covariate association scores for each permutation to create an empirical null distribution. A shuffled dataset should result in lower-scaled factor-covariate association scores compared to the original dataset, given that the cell labels are randomized after each permutation.

Empirical p-values were calculated as the number of permuted FCA scores that are as or more extreme than the observed association value, divided by the number of permutations. The number of significant associations for each covariate was determined using p-value < 0.05. We expect one significant factor association per covariate level in the ideal case. High values of this metric likely indicate false positive associations (low specificity) and zero values highlight false negative results (low sensitivity).

To evaluate model performance without relying on predefined thresholds (such as p-value = 0.05), we employed the Gini index, an inequality measure ranging from 0 to 1. A Gini index of 0 signifies perfect equality, where all values are identical, while a score of 1 indicates perfect inequality, with one value dominating the distribution. In our context, we aimed to assess the extent of association between factors and covariates across various levels of pre- and post-permutation. Following label shuffling, we expect the factor-covariate association scores to exhibit uniformity across all covariate levels (Gini score closer to 0). In cases of non-random baseline labels, sparse associations between factors and covariates would result in a Gini score closer to 1. To calculate the Gini index for a given FCA matrix, Gini was calculated for each covariate level separately and the average Gini index across all covariate levels is provided.

$$G_k = \frac{1}{2n^2 \bar{x}}\sum_{i=1}^{n}\sum_{j=1}^{n} |x_i - x_j|$$

Where $G_k$ is the Gini coefficient for a covariate level $k$, $n$ is the number of factors, and $\bar{x}$ are the mean FCA scores for a given covariate level. We then calculate the global Gini index as:

$$G = \frac{1}{K}\sum_{i=1}^{K} G_k$$

Where $K$ is the total number of covariate levels.

### Factor evaluation

We defined four categories of metrics: separability, effect size, specificity, and homogeneity to measure factor interpretability. Two are label-free (separability, effect size) and the other two are label-dependent (specificity, and homogeneity). Metrics for each category are organized into a table of factor interpretability scores (FIS), with metrics arranged as rows and factors as columns. Subsequently, the FIS undergoes row-wise scaling and is visualized as a heatmap to support the comparison of interpretability scores between factors.

sciRED uses the Silhouette score and bimodality index to assess separability. Homogeneity is measured using the arithmetic mean of the average scaled variance (ASV) table. The Simpson index is employed to evaluate factor specificity, while factor variance serves as an effect size measure (See Methods section "Correlation between interpretability metrics and overlap values").

### Separability (bimodality)

We use bimodality scores as an indicator for measuring the separability of factors. We use existing metrics developed for identifying genes with bimodal expression distributions, which are typically used to assess genes with prognostic value in discriminating patient subgroups. We selected metrics in accordance with Hellwig et al.'s comparative assessment of these scores based on survival times in a breast cancer dataset[72].

### Cluster-based bimodality measures

Cluster-based methods group observations into clusters, then calculate various statistics to measure the degree of distinctiveness of these clusters. Clusters were generated using a standard k-means[73] algorithm ($k = 2$) implemented using the scikit-learn[60] package. The following cluster-based bimodality measures are included in sciRED:

- Variance Ratio Score (VRS), also known as Calinski–Harabasz index (CHI)[74] assesses the proportion of variance reduction when splitting the data into two clusters. We can decompose the total sum of squares (TSS) to between-cluster sum of squares (BSS) and within-cluster sum of squares (WSS). VRS is then defined as the ratio of BSS and WSS.
- Weighted Variance Ratio score (WVRS) is similar to VRS but measures the variance reduction independent of the cluster sample sizes. In both cases, higher values indicate better separation between clusters and reflect bimodality.
- Silhouette score[75] measures the cohesion and separation of clusters, ranging from -1 to 1, where higher values indicate better-defined clusters. Silhouette is calculated based on mean intra-cluster distance (ICD) and the mean nearest-cluster distance (NCD). For each data point Silhouette is then given as $(NCD - ICD)/\max(ICD, NCD)$.
- Davies-Bouldin index[76] computes the average similarity between each cluster and its most similar cluster. It measures similarity as the ratio of within-cluster distances to between-cluster distances. The minimum value is zero with lower values indicating better

cluster separation. To ensure consistency with other metrics, we scale the inverse Davies-Bouldin Index.

Calinski–Harabasz, Silhouette, and Davies-Bouldin Indices were implemented using scikit-learn.

**Bimodality index.** The Bimodality Index[77] is another bimodality metric which was initially introduced to rank bimodal gene expression signatures within cancer gene expression datasets. We adopt the bimodality index to assess the bimodal nature of factor scores by modeling them as a mixture of two normal distributions using a Gaussian Mixture Model (GMM). Let $\mu_1$ and $\mu_2$ represent the means of the two normal distributions, and $\sigma$ the standard deviation. Given an equal variance assumption, the standardized distance $\delta$ between the distributions is calculated by:

$$\delta = \frac{|\mu_1 - \mu_2|}{\sigma}$$

Given GMM estimated parameters, we can calculate the proportion of observations in the first component $\pi$. We then compute the bimodality index (BI) as:

$$BI = \sqrt{\pi(1-\pi)} \times \delta$$

A higher BI indicates a stronger bimodal distribution, aiding in the identification of bimodal factor signatures. Scikit-learn was used to fit GMMs.

**Dip score.** Another bimodality evaluation metric is Hartigan's dip test[78], a statistical measure used to assess deviations from unimodality in distributions. It evaluates multimodality by comparing the maximum difference between the empirical distribution function and the unimodal distribution function that minimizes this difference. The dip score was computed for each factor, indicating the degree of bimodality present in the factor distribution. Higher dip scores suggest stronger evidence of bimodality, while lower scores indicate unimodality. Implementation was based on the diptest package.

### Effect size
The variance of factor scores across cells was employed as a measure of effect size consistency.

### Specificity
We assessed the Simpson diversity index and Shannon entropy as measures of specificity.

### Simpson diversity Index
The Simpson diversity index is a measure commonly used in ecology to quantify the diversity or evenness of species within a community[79]. It assesses the probability of encountering different species within a community and how evenly distributed these species are. We adopted the Simpson diversity index to measure specificity for individual factors within the context of FCA scores. By applying the Simpson index to each vector of FCA scores, we can evaluate the extent to which a factor uniquely characterizes a particular covariate level.

Mathematically, the Simpson diversity index $D$ is expressed as

$$D = \sum_{i=1}^{n} p_i^2$$

Where $p_i$ denotes the scaled score of factor-covariate level association (probability of a factor being chosen for a given covariate level $i$), and $n$ represents the total number of covariate levels. Here, the Simpson diversity index ranges between 0 and 1, where 0 indicates low factor specificity (maximum diversity, all factor association scores are equally distributed) and 1 indicates greater specificity (minimum diversity) of a factor towards a particular covariate level.

### Shannon diversity index
We applied Shannon entropy to measure the specificity of individual factors within FCA scores. By calculating Shannon entropy for each vector of FCA scores, we assess how uniquely a factor characterizes a particular covariate level.

Mathematically, Shannon entropy is defined as:

$$H = -\sum_{i=1}^{N} p_i \log(p_i)$$

where $p_i$ is the probability of a factor being associated with a given covariate level $i$, and $N$ is the total number of covariate levels. Shannon diversity index ranges from 0 (high factor specificity) to $\log(N)$ (low factor specificity).

### Homogeneity
To assess the homogeneity or even distribution of factor scores across different levels of a covariate, we compute the scaled variance for each covariate level. This metric quantifies the proportion of variance in the factor scores observed at a specific covariate level ($L$) relative to the total variance across all covariate levels[80]. Thus, the scaled variance $SV$ for a factor $x$ and a particular covariate level $L$ is computed as

$$SV = \frac{Var(x_L)}{Var(x)}$$

Here, $Var(x_L)$ denotes the variance of the factor scores $x$ corresponding to the covariate level $L$, $Var(x)$ represents the total variance of the factor scores across all cells.

To establish a unified metric across all levels of a single covariate (e.g., Batch1, Batch2, etc., representing levels of the covariate "Batch"), we adopt different approaches based on the covariate's number of unique levels. We compute both the geometric and arithmetic means of the scaled variances (SV) of all covariate levels for each factor. The arithmetic mean of SV values was chosen based on factor simulation results showing it performed best (explained below).

### Evaluating factor interpretability metrics using simulation
**Simulating mixture Gaussian distribution.** Factors were simulated under the assumption of being generated from a mixture of Gaussian distributions. Each Gaussian distribution represents the factor scores of the cells belonging to one covariate level. This simulation process was implemented using the following mathematical formulation:

Let $X$ be a random variable representing the simulated factors. We generated $n$ samples (cells) from a mixture of $K$ ($K=2$) normal distributions, each characterized by its mean $\mu_k$, standard deviation $\sigma_k$, and proportion $p_k$. The mixture distribution $X$ follows the distribution:

$$\sum_{k=1}^{K} p_k \cdot \mathcal{N}(\mu_k, \sigma_k^2)$$

where $\mathcal{N}(\mu_k, \sigma_k^2)$ denotes the probability density function (PDF) of the $k$-th normal distribution.

Given this assumption, 10 factors for 10000 cells were simulated for 100 rounds. $p_k$ was set to 0.5 for simplicity. $\mu$ and $\sigma$ were set by random sampling from uniform distributions (parameters: $\sigma_{min} = 0.5$, $\sigma_{max} = 1$, $\mu_{min} = 0$, $\mu_{max} = 4$).

### Correlation between interpretability metrics and overlap values
We assessed the efficacy of the proposed factor interpretability metrics by examining their correlation with the overlap between the two Gaussian distributions representing each factor. We anticipated that factors exhibiting greater overlap would demonstrate lower separability and specificity scores (negative correlation), along with

higher homogeneity values (positive correlation), and vice versa. Our goal was to identify metrics with higher absolute mean correlation values and lower variability, as illustrated in Fig. S21A and B. Based on simulation results (Fig. S21C), Silhouette and bimodality index indicate high performance for the separability measure. The arithmetic mean of the average scaled variance (ASV) table shows higher performance compared to the geometric mean for the homogeneity measure. Our simulation results indicated a superior performance of Simpson index compared to Shannon entropy[81] to measure factor specificity.

## Calculating overlap between double Gaussian distributions

The overlap between two Gaussian distributions with means $\mu_1$ and $\mu_2$, and standard deviations $\sigma_1$ and $\sigma_2$ respectively, is quantified based on the intersection of the PDFs of the two distributions. For distributions with unequal means and standard deviations, the overlap $O$ is computed as:

$$O = 1 - \frac{1}{2}\left[ erf\left(\frac{c - \mu_1}{\sqrt{2}\,\sigma_1}\right) - erf\left(\frac{c - \mu_2}{\sqrt{2}\,\sigma_2}\right)\right]$$

where $c$ represents the intersection point of the two PDFs and $erf$ is the error function.

## Pathway and gene set enrichment analysis

Gene set and pathway enrichment analysis methods were used to study the biological signatures represented by each factor. The gene scores corresponding to the factors of interest were selected from the loading matrix to order the list of genes from most to least contribution to the given factor. Pathway enrichment analysis was performed on the top 200 genes of the ordered loadings using the gprofiler2[82] (version 0.2.3) enrichment tool based on the default Gene Ontology Biological Process and Reactome gene set database sources, and using the "ranked" mode.

## Interpretation of factors using FCA and FIS tables

The FCA and FIS tables facilitate interpretation of identified factors. When covariates are available, evaluating the FCA table first can reveal factors associated with known covariates. Visual inspection of the FCA heatmap helps identify specific factor-covariate pairs, and automatic thresholding highlights factor-covariate pairs with scores that stand out against the background distribution. Ideally, factors of interest will exhibit high specificity, as summarized in the FIS table. Specificity measures how uniquely a factor associates with a particular covariate level (e.g., scRNA-seq technology type), with higher specificity enhancing interpretability.

The FIS heatmap includes three additional metric scores - bimodality (separability), effect size, and homogeneity - which capture distinct aspects of the factor distribution and help prioritize unexplained factors for further exploration. Bimodality, which is covariate-independent, reflects the distribution's modality, where high values indicate the presence of two distinct cell populations with high and low factor scores. Effect size, also covariate-independent, represents the distribution's spread and variance. Higher effect size indicates higher signal strength. Homogeneity is covariate-dependent and measures the relative spread of the data specific to a single covariate compared to the total spread. Homogeneity values closer to one suggest cells are well-mixed with respect to the covariate along the factor's axis. These metrics guide the identification of interpretable factors from a set of K factors generated by any factorization algorithm. However, factor distributions may show patterns beyond these metrics, and even one high metric value can justify follow-up interpretation, guided by enriched genes and pathways or expert knowledge.

## Modular design for flexible usage

sciRED's modular design allows integration with external data processing and normalization methods. Preprocessed data can be input directly into step 2 (factor identification). Additionally, if external matrix factorization has already been performed, the initial steps can be bypassed, with factors and covariates imported directly for factor-covariate association analysis.

## Data Preprocessing

The scMixology[17] dataset includes three human lung adenocarcinoma cell lines: HCC827, H1975, and H2228. These cell lines were cultured individually and subsequently processed. Single cells from each cell line were combined in equal proportions and libraries were generated using three protocols: CEL-seq2, Drop-seq, and 10x Genomics Chromium. The processed count data was obtained using the scPipe package in R and converted to .h5ad for import into Python. The data underwent log normalization and standardization using the "StandardScaler" function from scikit-learn package.

The stimulated PBMC data[26] includes 10x Genomics droplet-based scRNA-seq PBMC data from eight lupus patients before and after 6h-treatment with interferon-beta. Count data was extracted and analyzed using sciRED (number of components(k)=30) while modeling library size as a technical covariate. Three outlier cells were removed from the sciRED cell-by-factor score matrix, and factors F2 and F9 scores for 29,062 cells were visualized in Fig. 3.

The healthy human kidney map was constructed based on 19 living donors (10 female, 9 male)[28] including the transcriptomes of 27,677 cells. Filtered and normalized data was downloaded and analyzed using sciRED ($k = 30$).

The healthy rat total liver homogenate map includes four whole livers which were acquired from 8-10 week-old healthy male Dark Agouti and Lewis strain rats, and the resulting total liver homogenates went through two-step collagenase digestion and 10x Genomics droplet-based scRNA-seq[29]. Five outlier cells were removed from the score matrix, and factors F6 and F20 scores for 23,036 cells were visualized in Fig. 4. sciRED ($k = 30$) was applied to the count data while modeling library size as a technical covariate. Sample was not included as a technical covariate to preserve the strain-specific variations. SoupX[31] software (version: 1.6.2) was used to identify genes with the greatest contribution to the ambient RNA. We used the default automatic contamination fraction estimation (Rho) feature in the SoupX (autoEstCont function) to estimate Rho for each sample included in the total liver homogenate map of healthy rat livers. Subsequently, we extracted the estimated ambient RNA profile and identified the top 50 genes contributing the most to the ambient RNA in each sample. Genes were selected if they ranked among the high-scoring ambient RNA contributors in at least two samples. These selected genes were assessed for their presence among the strain-specific myeloid markers identified using both sciRED and standard differential expression methods. Differential expression analysis between the DA and LEW strains within the myeloid population (cluster 5) of the rat liver map was conducted using Seurat's FindMarkers function with default parameters (logfc.threshold = 0.1, min.pct = 0.01, min.cells.feature = 3, and min.cells.group = 3), implementing the non-parametric Wilcoxon rank-sum test.

The healthy human liver map[32] includes 8,444 parenchymal and non-parenchymal cells obtained from the fractionation of fresh hepatic tissue from five human livers. The liver tissue was obtained from livers procured from deceased donors deemed acceptable for liver transplantation. sciRED ($k = 30$) was applied to the filtered count while modeling library size as a technical covariate.

The human lung transplant dataset[27] includes donor lung biopsies from six transplant cases and over 108,000 cells. We performed sub-sampling on both genes and cells to systematically assess sciRED's runtime across different dataset sizes.

The two filtered 10k PBMC datasets, profiled using 10x Genomics single-cell 3' and 5' gene expression libraries, were downloaded from the 10x Genomics and directly analyzed using sciRED.

We applied sciRED to spatial transcriptomic data from Maynard et al. [83], focusing on identifying spatial gene expression patterns within the six-layered human dorsolateral prefrontal cortex (DLPFC). Specifically, we selected Visium samples from two subjects (Br8100 and Br5292), with four samples per subject. The data were subsetted by subject. sciRED's Poisson GLM was applied to adjust for library size.

### Reporting summary
Further information on research design is available in the Nature Portfolio Reporting Summary linked to this article.

## Data availability
The scMixology[17] dataset can be accessed from https://github.com/LuyiTian/sc_mixology. The stimulated PBMC datasets[26] were downloaded from muscData package (Kang18_8vs8) at https://github.com/HelenaLC/muscData. The healthy human kidney atlas[28] data files were downloaded from the UCSC Cell Browser at https://cells.ucsc.edu/?ds=living-donor-kidney. The processed healthy rat total liver homogenate map[29] was also downloaded from the UCSC Cell Browser at https://cells.ucsc.edu/?ds=rat-liver-atlas. The healthy human liver atlas[32] data was obtained from the R package HumanLiver, available at https://github.com/BaderLab/HumanLiver. The human lung transplant dataset[27] was downloaded from the cellxgene platform with GEO accession code GSE220797. The filtered 3' library PBMC dataset was downloaded from https://cf.10xgenomics.com/samples/cell-exp/4.0.0/Parent_NGSC3_DI_PBMC/Parent_NGSC3_DI_PBMC_filtered_feature_bc_matrix.h5. The 5' library PBMC data is available at https://cf.10xgenomics.com/samples/cell-vdj/5.0.0/sc5p_v2_hs_PBMC_10k/sc5p_v2_hs_PBMC_10k_filtered_feature_bc_matrix.h5. The human dorsolateral prefrontal cortex spatial transcriptomics data[83] was obtained using the spatialLIBD package. Source data are provided with this paper.

## Code availability
The Python package for sciRED, along with example case scenarios and the analysis code used in this manuscript, is freely accessible at https://github.com/BaderLab/sciRED. The code is archived and citable via Zenodo: Pouyabahar et al., sciRED, GitHub repository, https://doi.org/10.5281/zenodo.14593137 (2024).

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

## Acknowledgements
This research was supported by the University of Toronto's Medicine by Design initiative, which receives funding from the Canada First Research Excellence Fund (CFREF) to GDB; by the NRNB (U.S. National Institutes of Health, grant P41 GM103504) to GDB; by the Canadian Institutes for Health Research (grant PJT 469829 to GDB) and by the University of Toronto's Data Sciences Institute Doctoral Student fellowship program.

## Author contributions
Conceptualization, D.P., T.A., and G.D.B.; Methodology and software, D.P; Formal analysis, D.P., Writing – Original Draft, D.P.; Writing – Review & Editing, T.A., and G.D.B.; Supervision, G.D.B., T.A.; Funding Acquisition, G.D.B.

## Competing interests
GDB is an advisor for Deep Genomics and is on the Scientific Advisory Board of Adela Bio. No other competing interests are declared.
