## [Transparent Peer Review file · Nature Communications]

Interpretable single-cell factor decomposition using sciRED

Corresponding Author: Professor Gary Bader

Version 0:

Reviewer comments:

Reviewer #1

(Remarks to the Author)

Matrix factorization is a commonly used strategy for multi omics data analysis. The authors proposed a tool called sciRED for the factor analysis and interpretation of scRNA seq data. The authors also analyzed several scRNA seq data using sciRED to present the results. Specifically, the authors adopted several metrics for the interpretation and ranking of the factors. However, there are already many similar tools available for matrix factorization or decomposition of single-cell omics data, I am afraid that the novelty, performance, and scalability of this study need to be further confirmed.

1) SciRED is essentially a matrix factorization tool. At present, there are already a bunch of decomposition methods, such as references 4-16 in the manuscript, and others (Cleary, et al., 2017; Stein-O'Brien, et al., 2018). Although these methods may not provide quantitative metrics for interpretation of factors, the obtained factors from different methods can still be compared. But the manuscript lacks a comparison with existing methods. In the section on "sciRED improvement factor discovery", the authors only compared SciRED with PCA, which is clearly insufficient.

2) The authors point out that a major innovation of the study is the use of several metrics for factor interpretation. However, in other matrix factorization studies targeting scRNA-seq, the results are always explained and metrics such as contribution are used. For example, the study by Kotliar et al. (2019) also provided a very clear explanation of the results of matrix factorization. So, it is necessary for the authors to explain in more detail the differences between different metrics and how to automatically and comprehensively use these metrics for users' analysis.

3) Since the interpretation of factors is performed after matrix factorization, can those metrics proposed in this study be directly used on factors obtained from other matrix factorization methods?

4) The author used 4-5 data sets to demonstrate the application of sciRED, but only limited to scRNA-seq data. But as the author pointed out in the discussion, there are currently many omics data, such as spatial transcriptomics, scATAC seq, etc. To demonstrate the scalability and versatility of this tool, it is still necessary for the authors to demonstrate its application in other omics data.

5) Single cell data volume is usually large, especially when the data comes from multiple batches. The authors are suggested to test the performance of sciRED on data of different scales. Especially when the dataset is super large if integrated from multiple batches, it may need to do matrix factorization on individual batches due to computing capacity. Then, what is the difference between using sciRED on merged batches and combining factors from independently decomposing of individual batches? Then what is the difference between using sciRED on the merged batch and independently decomposing a single batch with sciRED before merging?

6) Single cell data is usually highly sparse, so how much does the sparsity of the data affect the decomposition results of sciRED?

7) The number of factors is one of the most critical parameters in most matrix factorization methods. But it seems that the authors did not describe in the manuscript how to determine the number of factors.

Cleary, B., et al. Efficient Generation of Transcriptomic Profiles by Random Composite Measurements. Cell 2017;171(6):1424-1436 e1418.

Kotliar, D., et al. Identifying gene expression programs of cell-type identity and cellular activity with single-cell RNA-Seq.

Elife 2019;8.

Stein-O'Brien, G.L., et al. Enter the Matrix: Factorization Uncovers Knowledge from Omics. Trends Genet. 2018;34(10):790-805.

(Remarks on code availability)

The authors provided installation instructions for the tool and two tutorials.

Reviewer #2

(Remarks to the Author)

Current gene program discovery methods need to annotate the resulting gene programs manually. The authors present a novel method (sciRED) to map identified factors to known biological variables and identify new gene programs. The applications of sciRED on multiple scRNA-seq datasets show the ability to identify sex- or condition-specific gene programs. One of the most interesting findings for me is the discovery of novel rare cell subtypes through the resulting factors.

Overall, the proposed framework is validated using multiple datasets. The framework can greatly improve our understanding of factorization in gene program discovery. While this study represents an exciting advancement in the field, I do have several concerns that need to be addressed:

Major:

1. The modeling of the gene expression assumes that technical covariates are identical across all genes, as the β is a $p \times 1$ vector. Could the authors elaborate on the rationale?
2. Fig. 1b is difficult to understand, as the elements in the box do not seem to be comparable. Further clarification would be beneficial.
3. In the PBMCs dataset, I am particularly interested in understanding the roles of factors that do not correspond to cell type or condition-specific activation. Could the authors provide more insight into this?
4. For the factor number, it looks like the authors used the same 30 in both the simple PBMCs dataset and the complex liver dataset. How was the factor number chosen? Is the model robust to different choices of the factor number?
5. Could the authors provide the results of other gene program discovery methods, such as Spectra, on the scMixology dataset? This comparison could further highlight the advantages of the proposed method.

Minor:

1. Typo correction
 - 1.1. Single-Cell Interpretable REsidual Decomposition (sciRED)
 - 1.2. "Factor-correlated," "scMixology" and "SCMixology" in the Introduction section. And the The first mention of "scMixology" should be accompanied by a citation.
 - 1.3. The full name of PCA should be provided upon its first mention.
 - 1.4. "PCA1" in "sciRED improves factor discovery"
 - 1.5. PCA factors should be PCs in Supp Fig. 2.
2. The first paragraph of the results should provide an overview of the method to enhance clarity.
3. The FCA table for all calculated factors in all datasets should be provided.

(Remarks on code availability)

The codes provide a detailed README file and example applications of the proposed method. I can install the package and run the code.

Reviewer #3

(Remarks to the Author)

Factor analysis can be useful for extracting meaningful factors that may be cell-type-specific or shared across cell types. This approach can complement clustering analysis and can be particularly valuable when the data do not naturally group into clusters or when clustering analysis fails to identify rare clusters. However, the results of factor analysis can be sensitive to parameter settings, and interpreting the factors may not always be straightforward. Thus, this paper presents an interesting approach to factor analysis and aids the interpretability, and I have some comments for the authors.

a) For the method, the authors first regress out the known confounding factors and then perform PCA on the residuals. This step may help extract subtle and meaningful factors. However, the residuals can be both positive and negative, which limits their use in other methods, such as non-negative matrix factorization (NMF), which is known to produce more interpretable factors than PCA. So I'm wondering why the authors did not simply use NMF and interpret the results as presented in the paper. Some of these factors may simply capture those confounding factors but we can simply ignore them.

b) For factor analysis models, the results can be sensitive to certain parameters, such as the number of genes used (including or removing some highly or lowly expressed genes), the number of factors analyzed, and data normalization methods. I did not see the details regarding the types of processing (e.g., removing some highly expressed genes etc) the authors conducted or the number of genes used. Including analyses that demonstrate how the results change with variations in the number of factors and the number of genes (e.g., including or removing highly expressed genes) used would

strengthen the paper.

c) The authors demonstrated that sciRED performs well, even in the presence of strong batch effects, using simple cell line data. However, they did not evaluate the performance of sciRED on more complex, real-world data with strong batch effects (e.g., from different human or mouse tissues, across various studies, or using different technologies). Including integrated analyses of complex data with strong batch effects could strengthen the analysis and the conclusions.

d) Data analysis is a repeated process and efficient algorithms can be very helpful. However, some of the components can be time-consuming for sciRED, e.g., permutation analysis and classifier-ensemble. It could be good to see some running time for the method on data with different numbers of cells (genes).

e) There seems to be some discrepancy in Fig. 5H, e.g., F28 captures the cell cycle in the text, but the figure uses F29.

f) The method has a Poisson GLM, which seems to work well for data with UMIs. Does the method work well for full-length data, e.g., Smart-seq2 data?

(Remarks on code availability)

I briefly checked the github repo, which is available for use and the authors also provide two tutorials for interested users.

Version 1:

Reviewer comments:

Reviewer #1

(Remarks to the Author)

The authors have addressed all my concerns.

(Remarks on code availability)

The code can be run successfully.

Reviewer #2

(Remarks to the Author)

The authors have addressed all of my concerns. The revised manuscript has better readability and the conclusions are clearer. I would like to recommend acceptance to Nature Communications.

(Remarks on code availability)

The code and tutorial are clearly presented.

Reviewer #3

(Remarks to the Author)

I'd like to thank the authors for adequately addressing my comments.

I don't have further questions.

(Remarks on code availability)

Code and tutorials are available from Github

Reviewer #1 (Remarks to the Author):

Matrix factorization is a commonly used strategy for multi omics data analysis. The authors proposed a tool called sciRED for the factor analysis and interpretation of scRNA seq data. The authors also analyzed several scRNA seq data using sciRED to present the results. Specifically, the authors adopted several metrics for the interpretation and ranking of the factors. However, there are already many similar tools available for matrix factorization or decomposition of single-cell omics data, I am afraid that the novelty, performance, and scalability of this study need to be further confirmed.

1) SciRED is essentially a matrix factorization tool. At present, there are already a bunch of decomposition methods, such as references 4-16 in the manuscript, and others (Cleary, et al., 2017; Stein-O'Brien, et al., 2018). Although these methods may not provide quantitative metrics for interpretation of factors, the obtained factors from different methods can still be compared. But the manuscript lacks a comparison with existing methods. In the section on “sciRED improvement factor discovery”, the authors only compared SciRED with PCA, which is clearly insufficient.

We thank the reviewer for their insightful comments. We expanded our comparisons to include three general factor decomposition methods—PCA, NMF, and ICA—alongside five state-of-the-art, single-cell-specific methods: Zinbwave, scCoGAPS, cNMF, Spectra, and scVI. These methods were benchmarked on two datasets: the scMixology dataset, which was used for comparison in our original manuscript, and stimulated PBMC datasets, which we added to the comparison analysis. These were chosen to represent distinct biological and technical contexts.

The scMixology dataset was selected as it represents a well-established, controlled scRNA-seq experiment with a reliable ground truth. This dataset includes scRNA-seq profiles of three human lung adenocarcinoma cell lines (HCC827, H1975, and H2228), each cultured separately and mixed in equal proportions, with libraries generated using three different protocols: CEL-seq2, Drop-seq, and 10x Genomics Chromium. In contrast, the stimulated PBMC dataset consists of 10x Genomics droplet-based scRNA-seq data from eight lupus patients, sampled before and after a 6-hour treatment with interferon (IFN)- β . This dataset provides a real-world example of single-cell data with multifactorial biological complexity.

We evaluated sciRED's factor discovery performance alongside the eight other methods across four comparison metrics: 1) the number of entangled covariates, representing covariates matched to multiple factors; 2) the number of known factors split across multiple covariates; 3) the number of covariate levels without an associated factor; and 4) run time. Lower values for these metrics indicate stronger performance. sciRED outperformed other methods in minimizing both entangled covariates and factors distributed across multiple covariates, with the exception of scCoGAPS on both datasets. All methods performed comparably in associating covariate levels with factors, especially for biological signals. Notably, however, scCoGAPS missed the megakaryocyte cell type in the PBMC dataset. The runtime analysis, presented on a log scale, demonstrates sciRED's scalability, while scCoGAPS and Zinbwave showed particularly slow performance, with Zinbwave unable to complete processing on the PBMC dataset on our computer due to scalability issues. All methods were run with default parameter settings on a workstation with an Intel 3.0GHz Xeon E5-2687W chip and 64 GB RAM. Spectra was applied exclusively to the PBMC dataset due to its reliance on the Cytopus knowledge base, which includes gene sets specific to standard scRNA-seq datasets but not to the cancer cell lines in the

scMixology dataset. We have updated the main results section to reflect these benchmarking analyses, and supplementary figures (Figures S2-S5) now provide visual comparisons of the metrics across methods.

To further explore sensitivity to the chosen factor number, we performed additional analyses on the scMixology dataset, setting the initial factor count to 30 and 10 (Figure S3). Some methods, particularly ICA and NMF, exhibited sensitivity to the factor number: ICA tended to split cell identities for cancer cell line H2228, and NMF for cancer cell line H1975 in scMixology, when the factor count was set to 30 whereas sciRED was robust to factor number.

As an additional point, sciRED's factor interpretation step applied to all methods after they are run on the data enables us to define useful comparison metrics that we used. These are new metrics that support the benchmarking of single cell genomics factor discovery methods and can be used by others in the future.

2) The authors point out that a major innovation of the study is the use of several metrics for factor interpretation. However, in other matrix factorization studies targeting scRNA-seq, the results are always explained, and metrics such as contribution are used. For example, the study by Kotliar et al. (2019) also provided a very clear explanation of the results of matrix factorization. So, it is necessary for the authors to explain in more detail the differences between different metrics and how to automatically and comprehensively use these metrics for users' analysis.

We apologize that this point was not clear. The interpretability metrics represent one component of sciRED's novel contributions. We also introduce a new pipeline that integrates varimax rotation with PCA, coupled with an automated consensus approach for identifying factor-covariate associations. We have adjusted the text to clarify.

We also updated the text to clarify the differences between metrics. The interpretability metrics in the FIS table assess specificity, bimodality (separability), effect size, and homogeneity. Specificity reflects how uniquely a factor associates with a particular covariate level (e.g. scRNA-seq technology type), with higher specificity supporting easier interpretability. Bimodality, effect size, and homogeneity capture distinct aspects of the factor distribution. Bimodality, which is covariate-independent, reflects the distribution's modality, where high values indicate the presence of two distinct cell populations with high and low factor scores. Effect size, also covariate-independent, represents the distribution's spread and variance. Higher effect size indicates higher signal strength. In contrast, homogeneity is covariate-dependent and measures the relative spread of the data specific to a single covariate compared to the total spread. Homogeneity values closer to one suggest cells are well-mixed with respect to the covariate along the factor's axis. These metrics guide the identification of interpretable factors from a set of K factors generated by any factorization algorithm. However, factor distributions may show patterns beyond these metrics, and even one high metric value can justify follow-up interpretation. In such cases, further interpretation should rely on enriched genes and pathways. We have added a methods section, "Interpretation of factors using FCA and FIS tables" to serve as a user guide for these metrics.

3) Since the interpretation of factors is performed after matrix factorization, can those metrics proposed in this study be directly used on factors obtained from other matrix factorization methods?

Yes. We thank the reviewer for this suggestion and have added a supplementary figure (Figure S10) to the manuscript to illustrate sciRED applied to NMF as an example. To address this question, we tested sciRED using NMF (replacing Step 1) on the PBMC dataset and evaluated the resulting factors with sciRED's standard pipeline. The FCA table indicates that many cell-type identity programs and simulation signals were captured using NMF. However, unlike factors derived from sciRED's default factor identification, NMF captured multiple covariates per factor, complicating interpretation. Ideally, each factor would correspond one-to-one with a single covariate.

Both FCA and FIS metrics can be applied in this context, similar to the standard pipeline. For instance, Factor 2, associated with megakaryocytes, shows high bimodality and effect size scores in the FIS table. Factors F5, F12, F13, and F23 also display higher specificity values, aligning them with specific cell types rather than a broad distribution across covariates. Figure S10F presents scatter plots of these four factors against F1, where each dot represents a cell colored by cell type, and box plots (Figure S10G) show factor distributions across cell type categories. Factors F4 and F21 exhibit the highest association scores for stimulation covariate levels, supported by low homogeneity scores for the stimulated covariate. Notably, F4 has the highest specificity score among the factors. Figures S10H and S10I show the distribution of cells along F1 vs. F4 and F21, colored by stimulation status, with box plots comparing factor scores between stimulation and control groups, as well as across cell types. Gene score analysis reveals immune-response-related genes among the top-scoring genes for these factors. Overall, these results indicate that sciRED's factor interpretation and evaluation steps can be directly adopted to guide the interpretation of factors identified through alternative matrix factorization methods.

4) The author used 4-5 data sets to demonstrate the application of sciRED, but only limited to scRNA-seq data. But as the author pointed out in the discussion, there are currently many omics data, such as spatial transcriptomics, scATAC seq, etc. To demonstrate the scalability and versatility of this tool, it is still necessary for the authors to demonstrate its application in other omics data.

We thank the reviewer for their helpful comment. We have added an example of sciRED's application to spatial transcriptomics which shows that sciRED can be successfully applied to this data type (Figures S11 and S12). Specifically, we used sciRED to analyze spatial transcriptomics data from Maynard et al. (2020), focusing on the six-layered human dorsolateral prefrontal cortex (DLPFC) to identify spatial gene expression patterns. This study uses the 10x Genomics Visium platform to define the spatial topography of gene expression in human DLPFC and identifies layer-enriched expression signatures, and is commonly used for spatial transcriptomics analysis method evaluation. The acquired tissue blocks span the six cortical layers and white matter (WM). Two pairs of spatial replicates were sampled from neurotypical adult donors. Each pair consists of two directly adjacent tissue sections. We applied sciRED to Visium samples selected from two subjects (Br8100 and Br5292). Each subject includes four samples in total. Data was downloaded using spatialLIBD package and then subset by subject.

Figure S11 shows sciRED's results on four samples from subject Br8100. We adjusted for library size using sciRED, followed by the factor decomposition step. The low correlation between factors and library size indicates that these factors are not influenced by technical variation related to total read count. The sciRED FCAT heatmap illustrates the association of factors with cortical layers, sample IDs, and replicate covariates. Factors 1, 2, 3, 5, and 10 capture distinct signatures related to cortical

layers and white matter (WM). The boxplot distributions for Factors 1, 2, 3, 5, and 10, along with projections of factor scores across each sample slide, are shown in Figures S11D and S11E-H. For instance, Factor 1 captures the white matter gene expression signature across all four samples, as indicated by both its boxplot distribution and its enrichment in the WM regions of all four Visium slides.

We also applied sciRED to samples from another subject, Br5292, showing that sciRED effectively captured gene expression signatures specific to dorsolateral prefrontal cortex layers. To assess whether removing only library size or both library size and sample IDs using a Poisson GLM would affect the identified factors, we performed a correlation analysis, revealing that both approaches yield equivalent factors. Factors 1, 3, 5, 6, and 10 capture distinct spatial gene expression patterns that can be clearly seen in the the projection of scores over the Visium slides in Figure S12D-G. This is also demonstrated by the boxplot distributions across cortical layers (Figure S12C).

5) Single cell data volume is usually large, especially when the data comes from multiple batches. The authors are suggested to test the performance of sciRED on data of different scales. Especially when the dataset is super large if integrated from multiple batches, it may need to do matrix factorization on individual batches due to computing capacity. Then, what is the difference between using sciRED on merged batches and combining factors from independently decomposing of individual batches? Then what is the difference between using sciRED on the merged batch and independently decomposing a single batch with sciRED before merging?

In our standard sciRED analysis pipeline, we do not recommend partitioning data by batch. Instead, all samples' raw count data should be merged, filtered to exclude genes and cells with zero total counts, and then processed through the GLM Poisson step. Partitioning the data is generally unnecessary, as our runtime analysis shows that sciRED scales linearly with both cell and gene counts (Figure S6). For example, sciRED analysis of a data set of more than 100,000 cells takes under 10 minutes to complete a single-threaded run on a workstation with an Intel 3.0GHz Xeon E5-2687W chip and 64 GB RAM.

For extremely large datasets, users can consider subsampling cells to include representative data from all batches or, alternatively, use a faster method for factor identification, which can then be assessed for biological interpretability using sciRED. Partitioning by batch may obscure biologically meaningful factors, particularly those representing signals shared across or unique to subsets of samples, which are often central to biological questions. Therefore, relying on subsampling across batches or using a scalable factor identification method provides a better alternative for handling extremely large datasets.

We have added a section titled "Data Pre-processing and Handling of Batch Effects with sciRED" to the manuscript to clarify these guidelines for multi-sample analysis.

6) Single cell data is usually highly sparse, so how much does the sparsity of the data affect the decomposition results of sciRED?

Sparsity is indeed a critical factor in single-cell transcriptomics, where data is inherently sparse. To assess how sparsity affects sciRED's decomposition results, we systematically varied the sparsity levels (0.01, 0.3, 0.5, 0.7, 0.9, 0.95, 0.99) in the Human Liver Atlas dataset by replacing increasing

proportions of gene expression values with zeros. We then evaluated the resulting factors based on average variance explained across all factors, maximum variance explained among factors, the percentage of matched factors, and the percentage of matched covariates at each sparsity level. Our results indicate that the average variance explained by all factors remains largely robust across different sparsity levels, likely due to sciRED's rotation step, as this trend is not observed for PCA factors. The maximum variance explained by factors is also more stable with sciRED's rotated factors compared to PCA as sparsity increases. As expected, the percentage of matched factors decreases with rising sparsity, with a marked drop around a sparsity threshold of ~70%. Figure S15 provides FCAT heatmaps at selected sparsity levels. At lower sparsity (0.01 and 0.5), FCA heatmaps show a robust distribution in terms of number of matched covariates per factor, while at higher sparsity (0.95 and 0.99), most covariates are matched to only one or a few factors, severely limiting interpretability. Typical sparsity is 40% to 90%, depending on the experimental procedure and study design³, which is well within the range where sciRED performs well. The relatively stable percentage of matched covariates across sparsity levels is due to the fact that, although many covariates are matched to a single factor, a match persists, thereby preventing a decline in the number of matched covariates.

7) The number of factors is one of the most critical parameters in most matrix factorization methods. But it seems that the authors did not describe in the manuscript how to determine the number of factors.

We thank the reviewer for this suggestion. We performed decomposition on the human liver map using 5, 10, 20, 30, and 50 factors (K) and compared the results. For sciRED, underestimating the number of factors results in identifying factors that capture multiple biological signals, whereas overestimating the factor count has minimal impact on performance, simply yielding additional factors with no covariate associations (Figure S16). Notably, sciRED was more robust to the number of factors than many competing methods such as NMF (Figure S3).

Specifically, Figure S16A shows the number of matched covariates for each factor count, while Figure S16B displays the number of matched factors at each covariate level. With a lower factor count, such as the 5-factor decomposition, higher per-factor covariate associations occur; for example, Factor 1 (F1) is matched with nine covariate levels. As the factor count increases, each factor aligns with fewer covariates, and the percentage of matched factors decreases gradually. By the 50-factor decomposition, 54% of the factors are no longer matched with any covariates.

As an example comparison, we correlated factors from K=10 and K=30 decompositions. In the K=10 decomposition, three factors are highlighted in the figure (Figure S16C): F1 (black), F3 (pink), and F7 (yellow). In the K=30 results, these correspond to distinct sets:

- F1 in K=10 is correlated with F1, F6, F8, F25, and F27 in K=30 (black box and asterisks).
- F3 in K=10 maps to F3, F13, and F16 in K=30 (pink box and asterisks).
- F7 in K=10 correlates with F2, F7, F15, F18, and F21 in K=30 (yellow box and asterisks).

This shows that with a stricter factor limit, each factor may capture multiple gene expression programs, complicating interpretation. The optimal factor count is dataset-specific, depending on biological and technical signal diversity. Although a Scree plot is standard for determining PCA factor counts, rotation disrupts ordering by explained variance. Empirically, a moderately high K (e.g., 30) has worked effectively in practice for various datasets. Starting with a larger K and excluding factors

unmatched with covariates or with low FIS scores is recommended as a general strategy yielding interpretable results.

We performed the same analysis on the PBMC dataset and observed similar results (Figure S17): as the number of factors increased, each factor aligned with fewer covariates, and a greater portion of factors remained unmatched at higher K values.

We've added a section titled "Impact of factor number (K) on decomposition results" to the methods section to clarify this point and provide guidance for users.

References:

Cleary, B., et al. Efficient Generation of Transcriptomic Profiles by Random Composite Measurements. *Cell* 2017;171(6):1424-1436 e1418.
Kotliar, D., et al. Identifying gene expression programs of cell-type identity and cellular activity with single-cell RNA-Seq. *Elife* 2019;8.
Stein-O'Brien, G.L., et al. Enter the Matrix: Factorization Uncovers Knowledge from Omics. *Trends Genet.* 2018;34(10):790-805.

Thank you for providing these helpful references.

Reviewer #1 (Remarks on code availability):

The authors provided installation instructions for the tool and two tutorials.

Reviewer #2 (Remarks to the Author):

Current gene program discovery methods need to annotate the resulting gene programs manually. The authors present a novel method (sciRED) to map identified factors to known biological variables and identify new gene programs. The applications of sciRED on multiple scRNA-seq datasets show the ability to identify sex- or condition-specific gene programs. One of the most interesting findings for me is the discovery of novel rare cell subtypes through the resulting factors.

Overall, the proposed framework is validated using multiple datasets. The framework can greatly improve our understanding of factorization in gene program discovery. While this study represents an exciting advancement in the field, I do have several concerns that need to be addressed:

Major:

1. The modeling of the gene expression assumes that technical covariates are identical across all genes, as the β is a $p \times 1$ vector. Could the authors elaborate on the rationale?

We thank the reviewer for catching this typo. The model actually assumes that technical covariate coefficients are represented by a matrix of size $p \times g$, where g is the number of variables or genes and p is the number of covariates. This formulation allows each technical covariate to have gene-specific effects. We have corrected this in the manuscript.

2. Fig. 1b is difficult to understand, as the elements in the box do not seem to be comparable. Further clarification would be beneficial.

We appreciate the reviewer's feedback and have updated Figure 1b to improve clarity. In the revised figure, models and algorithms are now color-coded in blue, while input, output, and intermediate data elements are in white boxes. Additionally, we reformatted the pipeline to represent technical covariates as an input rather than an intermediate step, which more accurately reflects sciRED's design.

3. In the PBMCs dataset, I am particularly interested in understanding the roles of factors that do not correspond to cell type or condition-specific activation. Could the authors provide more insight into this?

Thank you for this question. We performed an in-depth analysis on the factors from the PBMC dataset that did not align with specific cell types or condition-related activation (presented in Reviewer Figure 1, see end of rebuttal). Out of the 30 factors identified, eleven (F1, F3, F4, F8, F13, F15, F16, F17, F19, F21, F22) showed distinct biological signatures, while factors with indices above 22 (F23-30) appeared to lack meaningful signal. To visualize these results, we provided a dot plot showing the top 20 genes with the highest positive and negative loadings for each factor. Genes with positive loadings are highlighted in red, while those with negative loadings appear in green. Interpreting the sign of these factor scores requires consideration of cell-type enrichments, as the signs alone are not meaningful. For example, Factor 3 is negatively enriched in a subset of CD4 T cells, with highly negatively loaded genes such as HSPB1 and HSPE1, which are abundantly expressed in this subpopulation. For pathway analysis, we used either the top 200 genes with the highest positive or negative loadings, depending on factor distribution across cell types. Pathways based on negatively loaded genes are colored green in the bar plots, while those based on positively loaded genes are colored red.

Our findings suggest that many of these factors (F1, F15, F16, F19, F21, F22) capture immune response signatures in monocytes. Below is a summary of the primary interpretations for each significant factor:

F1: Enrichment of cytoskeleton organization in dendritic cells and monocytes.

F3: ER processes and protein folding in CD4 T cells, with heat shock proteins likely indicating cellular stress in T cells.

F4: Ribosomal translation in lymphocytes.

F8: Enrichment of protein production pathways in B cells, potentially representing antigen-presenting B cells or progenitors.

F13: Regulation of T cell activation and proliferation shared between CD4 and CD8 T cells.

F15: Enrichment in chemotactic chemokines, possibly representing inflammatory monocytes.

F16: Innate immunity and inflammatory response in monocytes.

F17: Presence of GZMB, IRF8, and IRF7, suggesting an inflammatory dendritic cell subpopulation.

F19: Oxidative stress and apoptotic signaling in monocytes, marked by SOD2.

F21: Immune response regulation (e.g., CD9, TGFB1) in monocytes and dendritic cells.

F22: Inflammatory response in CD14 monocytes.

4. For the factor number, it looks like the authors used the same 30 in both the simple PBMCs dataset and the complex liver dataset. How was the factor number chosen? Is the model robust to different choices of the factor number?

The number of factors (K) significantly impacts the interpretability of results, so we have added further explanation in the manuscript, including a new section “Impact of factor number (K) on decomposition results”, and provided supplementary figures (Figures S16 and S17) to guide users. Additionally, please refer to our response to Reviewer #1, Question 7, for a detailed discussion of this analysis.

To evaluate how factor count affects decomposition results, we tested factor counts of 5, 10, 20, 30, and 50 (K) on the human liver dataset. Figures S16A and S16B in the supplementary material demonstrate two key patterns: (1) the number of matched covariates per factor, and (2) the distribution of matched factors across covariates as K varies. When a lower K (e.g., 5) is used, each factor matches with more covariates (e.g., F1 matches with nine covariates in the 5-factor decomposition), which suggests that multiple biological signals are condensed within a single factor. Conversely, as K increases, each factor aligns with fewer covariates, but the overall percentage of matched factors declines. For instance, at K=50, 54% of factors no longer align with any covariates.

For a more granular comparison, we correlated factors from K=10 and K=30. This revealed that some factors from K=10 represent composite signals split into distinct factors at K=30. For example, F1 in K=10 correlates with F1, F6, F8, F25, and F27 in K=30, indicating that a lower factor count might aggregate diverse signals, complicating interpretation. While Scree plots are often used to determine factor count in PCA, rotation disrupts sorting by the explained variance in sciRED. Therefore, starting with a moderately high K, such as 30, tends to improve interpretability and factor specificity.

We conducted a similar analysis on the PBMC dataset, observing consistent trends: as K increases, each factor aligns with fewer covariates, and the proportion of unmatched factors rises (Figure S17).

Overall, for sciRED, underestimating the number of factors results in identifying factors that capture multiple biological signals, whereas overestimating the factor count has minimal impact on performance, simply yielding additional factors with no covariate associations. Notably, sciRED is more robust to the number of factors than many competing methods such as NMF (Figure S3).

5. Could the authors provide the results of other gene program discovery methods, such as Spectra, on the scMixology dataset? This comparison could further highlight the advantages of the proposed method.

We expanded our comparisons in the manuscript to include three general factor decomposition methods—PCA, NMF, and ICA—alongside five state-of-the-art, single-cell-specific methods: Zinwave, scCoGAPS, cNMF, Spectra, and scVI. These methods were benchmarked on two datasets: the scMixology and stimulated PBMC datasets, chosen to represent distinct biological and technical contexts.

Please see our response to reviewer 1, comment 1 for more details.

Minor:

1. Typo correction

1.1. Single-Cell Interpretable REsidual Decomposition (sciRED)

1.2. “Factor-correlated,” “scMixology” and “SCMixology” in the Introduction section. And the The first mention of “scMixology” should be accompanied by a citation.

1.3. The full name of PCA should be provided upon its first mention.

1.4. “PCA1” in “sciRED improves factor discovery”

1.5. PCA factors should be PCs in Supp Fig. 2.

We thank the reviewer for their careful evaluation and for helping to improve the clarity of the text. These changes have been incorporated into the manuscript as suggested.

2. The first paragraph of the results should provide an overview of the method to enhance clarity. Done.

3. The FCA table for all calculated factors in all datasets should be provided.

The complete factor-covariate association (FCA) tables for all datasets have been added to the manuscript for clarity. Specifically, the FCA table for the healthy kidney atlas is included in Figure 3. For the scMixology dataset, the sciRED FCA table is provided in the benchmarking Figure S3. Similarly, the FCA table for the PBMC dataset is available in both the benchmarking figures (Figure S5) and in the main Figure 3. Main Figure 5 includes the full FCA table for the human liver map, and the missing FCA table for the rat liver map has now been added as Figure S8A.

Reviewer #2 (Remarks on code availability):

The codes provide a detailed README file and example applications of the proposed method. I can install the package and run the code.

Reviewer #3 (Remarks to the Author):

Factor analysis can be useful for extracting meaningful factors that may be cell-type-specific or shared across cell types. This approach can complement clustering analysis and can be particularly valuable when the data do not naturally group into clusters or when clustering analysis fails to identify rare clusters. However, the results of factor analysis can be sensitive to parameter settings, and interpreting the factors may not always be straightforward. Thus, this paper presents an interesting approach to factor analysis and aids the interpretability, and I have some comments for the authors.

a) For the method, the authors first regress out the known confounding factors and then perform PCA on the residuals. This step may help extract subtle and meaningful factors. However, the residuals can be both positive and negative, which limits their use in other methods, such as non-negative matrix factorization (NMF), which is known to produce more interpretable factors than PCA. So I'm wondering why the authors did not simply use NMF and interpret the results as presented in the paper. Some of these factors may simply capture those confounding factors but we can simply ignore them.

As detailed in our response to Reviewer #1, Question 3, we performed factorization on the PBMC dataset using NMF in place of sciRED's default factor identification (PCA followed by rotation), and then evaluated the resulting factors. Our analysis shows that sciRED outperforms NMF. NMF captures multiple covariates within single factors in our tests, which, while useful for identifying broad covariate influences, led to less interpretable factors compared to sciRED's default configuration, where factors ideally align one-to-one with distinct covariates. Changing the factor number with NMF is not guaranteed to solve this issue (Figure S3, shown with scMixology data). While the FCA table is compatible with NMF factors and provides useful insights (e.g., identifying factors representing cell type identities along with immune response gene signatures), NMF's broader covariate associations

per factor made it challenging to isolate unique signals, and sciRED's initial factorization step indicates higher performance in isolating distinct biological and technical signals. We've added these results as Figure S10 to the manuscript.

We also expanded our comparisons to include three general factor decomposition methods—PCA, NMF, and ICA—as well as five state-of-the-art, single-cell-specific methods: Zinbwave, scCoGAPS, cNMF, Spectra, and scVI. In brief, our benchmarking indicates that sciRED generally outperforms other methods in terms of combined interpretability and runtime efficiency. Further details are provided in the main text and Figures S2-S5.

b) For factor analysis models, the results can be sensitive to certain parameters, such as the number of genes used (including or removing some highly or lowly expressed genes), the number of factors analyzed, and data normalization methods. I did not see the details regarding the types of processing (e.g., removing some highly expressed genes etc) the authors conducted or the number of genes used. Including analyses that demonstrate how the results change with variations in the number of factors and the number of genes (e.g., including or removing highly expressed genes) used would strengthen the paper.

We've added a methods section, "Data pre-processing and handling of batch effects with sciRED" to provide users with further details on these parameters and data preprocessing in general.

For normalization, sciRED directly uses the count matrix as input without relying on external normalization. Instead, we incorporate library size as a covariate within the Poisson GLM, which normalizes for total read counts per cell.

For gene filtering, we applied variance and expression thresholds to select informative genes, excluding lowly-expressed and low-variance genes that could obscure signal detection. This is consistent with many single-cell analysis workflows. Initially, we remove all genes with zero expression and, by default, select the top 2000 highly variable genes (HVGs). To assess the effect of using HVGs versus all genes, we ran sciRED on the stimulated PBMC dataset with both HVGs and the complete gene set. The FCA tables for each setting are provided in the Internal Figures 2A and 2B. Correlation analysis (Internal Figure 2C) indicates that key signals—such as cell type identity and stimulation effects—are identified in both cases. The primary distinction is runtime: using HVGs completes in 2.21 minutes while analyzing the entire genome takes 21.53 minutes on the same computer. Given these comparable results, we recommend using highly variable genes for enhanced run-time efficiency. See Reviewer Figure 2 (see end of rebuttal).

Regarding factor count, please refer to our response to Reviewer #1, Question 7, and the associated supplementary analysis. In this analysis, we examine results across different factor numbers ($K = 5, 10, 20, 30,$ and 50), highlighting how factor alignment with covariates shifts as K increases and offering guidance on optimal factor selection. Briefly, too few factors reduced performance, but too many factors had minimal effect on performance for sciRED unlike for NMF where too many factors led to poorer performance (Figure S3). We have updated the manuscript to include these results and guidance for users in terms of parameter selection.

c) The authors demonstrated that sciRED performs well, even in the presence of strong batch effects, using simple cell line data. However, they did not evaluate the performance of sciRED on more

complex, real-world data with strong batch effects (e.g., from different human or mouse tissues, across various studies, or using different technologies). Including integrated analyses of complex data with strong batch effects could strengthen the analysis and the conclusions.

Thank you for this suggestion. To further evaluate sciRED's performance in handling strong batch effects, we applied it to two PBMC datasets profiled using 10x Genomics single cell 3' and 5' gene expression libraries, where batch effects from different assays are prominent (Figure S14). The UMAP projection of these concatenated datasets confirms the presence of strong batch effects, as cells do not integrate well without correction (Figures S14AB). We applied sciRED on the combined count matrices, regressing out the library size and sample IDs as covariates in the Poisson GLM step (no external batch correction method applied). Figures S14C and S14D display the resulting FCA and FIS tables, showing that sciRED effectively captures cell type identity programs without influence from the assay type. For instance, Factors F1 and F4, which identify CD14+ monocytes and B cells respectively, are well integrated across both assays. The distributions of cells over F1 and F4, colored by cell type and assay, demonstrate effective batch integration, and the corresponding box plots, and the enrichment patterns of these factors over UMAP further confirm these results. Together, these findings show that sciRED can reliably capture biological signals in the presence of strong batch effects without requiring additional external integration methods. Additionally, we have added further details on these analyses in the methods section for user clarity.

d) Data analysis is a repeated process and efficient algorithms can be very helpful. However, some of the components can be time-consuming for sciRED, e.g., permutation analysis and classifier-ensemble. It could be good to see some running time for the method on data with different numbers of cells (genes).

Thank you for highlighting this important point. The permutation analysis is not intended or provided for user application; it is only used once for benchmarking in the second step of sciRED to identify the classifiers that best match factors with covariate levels. The selected classifiers are now the standard ones used in the sciRED method. To evaluate sciRED's runtime relative to the number of cells and genes, we downloaded the Human Lung Transplants dataset from cellxgene, which includes donor lung biopsies from six transplant cases and over 108,000 cells. We then performed subsampling on both genes and cells to systematically assess sciRED's runtime across different dataset sizes.

Specifically, we tested:

- Number of cells (with genes fixed at 2000): 100K, 80K, 60K, 40K, and 20K.
- Number of highly variable genes (with cells fixed at 40K): 500, 2000, and 5000.

The runtime for each analysis is shown in Figure S6. As the scatter plots indicate, sciRED's runtime scales linearly with both the number of cells and the number of genes. For instance, sciRED takes under 10 minutes to complete a single-threaded run on a workstation with an Intel 3.0GHz Xeon E5-2687W chip and 64 GB RAM. We have included this runtime analysis in the manuscript and clarified the methods section to better explain the use of permutation analysis for benchmarking.

e) There seems to be some discrepancy in Fig. 5H, e.g., F28 captures the cell cycle in the text, but the figure uses F29.

We thank the reviewer for identifying this discrepancy, this was an error and has now been corrected in the text that it is F29 that captures the cell cycle.

f) The method has a Poisson GLM, which seems to work well for data with UMIs. Does the method work well for full-length data, e.g., Smart-seq2 data?

Currently, we do not recommend using the Poisson GLM step in sciRED for full-length, plate-based datasets such as Smart-seq2, as these data typically follow a different statistical distribution compared to UMI-based data. Specifically, Smart-seq2 data tend to have larger cell library sizes, detect more genes per cell, and exhibit higher zero-inflation rates than UMI-based methods like Drop-seq or 10x Genomics (Jiang et al. 2022). These differences make zero-inflated models (e.g., zero-inflated Poisson or zero-inflated negative binomial) more appropriate for Smart-seq2 data. In contrast, UMI-based datasets are generally well-modeled by non-zero-inflated distributions (e.g., Poisson), aligning with sciRED's current GLM framework.

Users can apply their custom preprocessing approach and then input these data into sciRED's factor identification which would allow sciRED to be used on Smart-seq2 data. Similar to an analysis using NMF-derived factors, by generating factors from Smart-seq2 data using zero-inflated models, users could then input these factors into sciRED for downstream analysis. We now mention this in the discussion.

Reviewer #3 (Remarks on code availability):

I briefly checked the github repo, which is available for use and the authors also provide two tutorials for interested users.

References

1. Crowell, H. L., Morillo Leonardo, S. X., Sonesson, C. & Robinson, M. D. The shaky foundations of simulating single-cell RNA sequencing data. *Genome Biol.* **24**, 62 (2023).
2. Cao, Y., Yang, P. & Yang, J. Y. H. A benchmark study of simulation methods for single-cell RNA sequencing data. *Nat. Commun.* **12**, 6911 (2021).
3. Baruzzo, G., Patuzzi, I. & Di Camillo, B. SPARSim single cell: a count data simulator for scRNA-seq data. *Bioinformatics* **36**, 1468–1475 (2020).

Reviewer Figure 1: Evaluation of PBMC Factors Unmatched with Pre-defined Biological Covariates. A) FCA heatmap; B) FIS heatmap; C) correlation analysis between unmatched factors and technical covariates (total counts and total features). Dot plots display the top 20 genes with the highest positive and negative loadings for factors D) F1, E) F3, F) F4, G) F8, H) F13, I) F15, J) F16, K) F17, L) F19, M) F21, and N) F22. Genes with positive loadings are highlighted in red and those with negative loadings in green. Boxplots (center) show the distribution of factor scores across various cell types. Pathway bar plots (right) highlight the top enriched GO terms based on the top 200 genes with the highest positive or negative loadings, as distributed across cell types. Pathways from negatively loaded genes are shown in green, while those from positively loaded genes are shown in red.

A**Highly Variable Genes**#genes: 2000
#cells: 29065Run time132.49 seconds
2.21 minutes**B****All Genes Included (post - QC)**#genes: 18890
#cells: 29065Run time1291.57 seconds
21.53 minutes**C****Pairwise Correlation Between Factors from
PBMC dataset with all genes and HVGs alone**
Reviewer Figure 2: Impact of gene filtering on sciRED's decomposition. We applied sciRED to the stimulated PBMC dataset using both highly variable genes (HVGs) and the complete gene set. A) FCA table for the analysis with HVGs. B) FCA results for the analysis with all genes included. Running sciRED with HVGs completes in 2.21 minutes, whereas using the full gene set takes 21.53 minutes. C) Correlation heatmap between the factors from the two analyses (rows: all genes; columns: HVGs) demonstrates that key signals, such as cell type identity and stimulation effects (indicated by arrows), are preserved across both analyses.